# Are Deep Speech Denoising Models Robust to Adversarial Noise?

**Will Schwarzer**[*,1]**, Philip S. Thomas**[1]**, Andrea Fanelli**[2]**, Xiaoyu Liu**[†,3]
[1]University of Massachusetts, [2]Dolby Laboratories, [3]Meta

## Abstract

Deep noise suppression (DNS) models enjoy widespread use throughout a variety of high-stakes speech applications. However, we show that four recent DNS models can each be reduced to outputting unintelligible gibberish through the addition of psychoacoustically hidden adversarial noise, even in low-background-noise and simulated over-the-air settings. For three of the models, a small transcription study with audio and multimedia experts confirms unintelligibility of the attacked audio; simultaneously, an ABX study shows that the adversarial noise is generally imperceptible, with some variance between participants and samples. While we also establish several negative results around targeted attacks and model transfer, our results nevertheless highlight the need for practical countermeasures before open-source DNS systems can be used in safety-critical applications.

## 1 Introduction

Deep neural networks (DNNs) have found widespread use in speech denoising tasks (herein considered synonymous with *speech enhancement* and *noise suppression*). With existing usage in video-conferencing software (Cutler, 2022) and speech recognition systems (Milling et al., 2024), and potential future usage in hearing aids (Westhausen et al., 2024), the robustness of such deep noise suppression (DNS) models is clearly of paramount concern.

However, it is well-documented that DNNs are often susceptible to adversarial perturbations—slight modifications to the input data that are subtle or imperceptible to humans, but which can lead to dramatically incorrect outputs from DNNs (Szegedy et al., 2014). This vulnerability has been extensively studied in domains such as automatic speech recognition (ASR) (Carlini & Wagner, 2018; Schönherr et al., 2018; Qin et al., 2019) and speaker recognition (Gong & Poellabauer, 2017; Wang et al., 2020), where attacks induce the models to mistranscribe speech or misclassify speakers.

Given their ubiquity and the vulnerability of similar models, we posit that DNS models are appealing targets for adversarial attacks. Beyond videoconferencing and hearing aids, neural speech enhancement is actively studied for mobile telephony (Tan et al., 2021), emergency-responder communications (Brodersen et al., 2019), and air-traffic-control radio (Wu et al., 2024; Yu et al., 2024). Many of these systems rely on open-source models with publicly available weights, giving a potential attacker full gradient access. Moreover, safety-critical voice channels often carry stereotyped, high-stakes phrases (evacuation orders, controller clearances, distress calls), so an attacker need only target a small set of known utterances to cause harm. Yet this threat might appear limited: DNS models are *designed* to remove noise, and prior attacks on speech enhancement models were perceptible, limited to high-noise settings, and might not work over-the-air (Dong et al., 2023).

To the contrary, we show that, across a variety of settings, **psychoacoustically hidden noise can cause four recent DNS models to output unintelligible gibberish**, thus raising urgent security concerns for such open-source DNS models (Defossez et al., 2020; Chen et al., 2022; Zhao et al., 2022; Lu et al., 2024). The success of our attacks varies little by setting, such as the presence and strength of reverb and normal background noise, and we demonstrate attacks in simulated over-the-air settings. While our attacks are white-box, and we find that naive model transfer fails, this is cold comfort: gradients are always available in open-source DNS models.

---

[*]Correspondence to: `wschwarzer@umass.edu`
[†]Work done while at Dolby Laboratories.

We also identify some areas for optimism. First, attacks appear to work best when designed for only a single utterance from the speaker; similar to other audio tasks (Neekhara et al., 2019; Abdoli et al., 2019; Zhang et al., 2021; Sun et al., 2024), imperceptible universal perturbations (Moosavi-Dezfooli et al., 2017) are not yet possible in this domain. Second, we find that one of the models we test, Full-SubNet+ (Chen et al., 2022), enjoys limited protection from white-box attack due to exploding gradients, though previous research suggests this is easily circumvented (Athalye et al., 2018). Third, *targeted* attacks may succeed according to objective metrics without subjectively producing the target utterance. Finally, simple Gaussian perturbation appears to be a moderately effective baseline defense; nevertheless, the prior success of adaptive attacks (Tramer et al., 2020) highlights that safety-critical DNS systems will need more sophisticated defenses.[1]

**Contributions.**

1. *Systematic study of imperceptible adversarial attacks on speech denoising models.* Contrary to prior image–denoising results, we show that four state-of-the-art DNS models can be driven to produce unintelligible outputs by psychoacoustically hidden perturbations, in speech settings ranging from nearly-clean ($70\,\mathrm{dB}$ SNR, no reverb) to noisy and reverberant. We also show that auditory masking offsets allow easy tuning of power versus imperceptibility (Appendix D.3).

2. *Evidence: Human study, comprehensive computational measurement, and samples.* We demonstrate the success of our core untargeted attacks using three complementary approaches: **a)** transcription and ABX studies with audio/multimedia experts; **b)** five distinct computational metrics; **c)** online samples to allow the reader to evaluate subjective imperceptibility and attack success.

3. *Masking- and RIR-aware attack framework.* We show that clipping to auditory masking thresholds in STFT space offers a ready-made projection operator for projected gradient descent (Madry et al., 2018). To ensure imperceptibility in simulated over-the-air experiments where the perturbation itself is convolved with a non-invertible room-impulse response, we evaluate a combination of Wiener deconvolution and gradient descent-based projection.

4. *Mechanistic insights: Gradient flow matters more than model size and input features.* We show that, contrary to common wisdom (Madry et al., 2018), the size of the models we test matters little for robustness; instead, the only protection we find comes from obfuscated gradients in Full-SubNet+ (Chen et al., 2022), protection which is known to be brittle (Athalye et al., 2018).

5. *Practical threat analysis.* Attacks are model- and utterance-specific and require gradient access, but successful over-the-air white-box attacks still preclude the deployment of open-source models in safety-critical applications such as hearing aids without additional defenses. We also quantify the partial protection offered by Gaussian perturbation as a baseline defense.

## 2 RELATED WORK

**Adversarial perturbations for audio models.** Adversarial perturbations originated in image classification (Szegedy et al., 2014), and have since been studied in audio tasks. Like visual attacks, audio perturbations are most commonly studied in classification tasks such as speaker recognition (Gong & Poellabauer, 2017; Zhang et al., 2020; Wang et al., 2020; Chen et al., 2021; Jati et al., 2021; Xie et al., 2021; Shamsabadi et al., 2021; Chen et al., 2023; Li et al., 2020), speaker verification (Kreuk et al., 2018; Zhang et al., 2021), environmental sound classification (Abdoli et al., 2019; Xie et al., 2021; Ntalampiras, 2022), and speech command recognition (Zhang et al., 2017; Xie et al., 2021).

Successful attacks also target audio models with discrete outputs, like ASR systems (Cisse et al., 2017; Alzantot et al., 2018; Carlini & Wagner, 2018; Schönherr et al., 2018; Qin et al., 2019; Neekhara et al., 2019; Sun et al., 2024; Jin et al., 2024). This work includes demonstrating imperceptible targeted attacks that force ASR systems to output attacker-chosen transcriptions.

The success of attacks on complex tasks like ASR raises questions about the vulnerability of models with continuous audio outputs. However, research on attacking generative audio models is limited. Takahashi et al. (2021) and Trinh (2022) demonstrated vulnerability in speech separation models, though the required perturbation perceptibility remains unclear. Dong et al. (2023) first attacked speech-enhancement networks, but their study left open questions regarding: **(i) Imperceptibility.**

---

[1]Samples and code are available online.

They used $\infty$-norm-bounded, audible perturbations; we use strict psychoacoustic masking (with pre/post masking and a $-12$ to $-16\,\mathrm{dB}$ offset) for imperceptible samples. **(ii) Attack goals.** Their attacks were untargeted; we analyze both intelligibility destruction and *targeted* utterance injection. **(iii) Real-world coverage.** Their experiments covered limited conditions ($[-8, 8]\,\mathrm{dB}$ SNR, no reverb, two models). We test four open-source DNS models in diverse conditions (near-clean to noisy, with/without reverb, simulated over-the-air), and report on transferability, universal perturbations, and defenses. Our work thus generalizes and significantly extends the threat landscape they outlined.

**Imperceptible audio perturbations.** Prior work highlights the complexity of ensuring imperceptible audio perturbations. Standard image perturbations are constrained by p-norms (initially 2-norm (Szegedy et al., 2014; Fawzi et al., 2018), now often $\infty$-norm (Goodfellow et al., 2015; Moosavi-Dezfooli et al., 2017)). User studies suggest these norm constraints yield imperceptible perturbations for classification tasks like speaker verification (Kreuk et al., 2018), and for weaker ASR attacks (e.g., untargeted, single-word attacks (Cisse et al., 2017; Alzantot et al., 2018)).

However, recent work indicates norm-constrained perturbations may lack capacity for stronger sentence-level targeted ASR attacks (Schönherr et al., 2018; Qin et al., 2019), prompting the use of psychoacoustic models for masking. Yet, even psychoacoustically masked perturbations can be shown perceptible in user studies, implying that difficult attacks still push perceptually solid constraints to their limits. For our attacks, we therefore enhance existing models (§4.2).

## 3 BACKGROUND

We study the existence of *adversarial perturbations* for DNS models. As background, we first review the standard behavior of DNS models and important characteristics of the models we study. We then formally state the definition of adversarial perturbations for DNS models, and review auditory masking, a perceptibility constraint for audio attacks (Schönherr et al., 2018; Qin et al., 2019).

### 3.1 SPEECH DENOISING

The primary goal of speech denoising models is to remove background noise from a speech signal (Defossez et al., 2020; Chen et al., 2022; Zhao et al., 2022; Lu et al., 2023; 2024). In some cases (Defossez et al., 2020), but not all (Chen et al., 2022; Zhao et al., 2022; Lu et al., 2023), the model is also trained to dereverberate signals that have been distorted by the environment or microphone.

Concretely, in this paper, speech denoising models are functions $f : \mathbb{R}^n \to \mathbb{R}^n$ that map to and from the space of audio waveforms. We model the denoising model's input as some clean speech $y \in \mathbb{R}^n$, mixed additively with some (possibly zero) background noise $b \in \mathbb{R}^n$, and (optionally) convolved with some room impulse response (RIR) $r \in \mathbb{R}^m$: $x = r * (y + b)$. (Note that some work only convolves the speech with $r$ (Zhao et al., 2022); in this paper, we convolve both speech and background noise.) Given this input, the model produces $f(x) = \hat{y}$, where $\hat{y}$ is as close to either $y$ or $r * y$ as possible. For consistency with past benchmarks (Dubey et al., 2024), and because we do not attempt to evaluate healthy model behavior, we assume that the model attempts to infer $y$.

#### 3.1.1 MODELS

We test the robustness of four recent open-source DNS models with publicly available checkpoints: Demucs, published online as Denoiser (Defossez et al., 2020); Full-SubNet+ (Chen et al., 2022); FRCRN (Zhao et al., 2022); and MP-SENet (Lu et al., 2023; 2024). The models we study operate in either the time domain (waveforms) or the time-frequency (TF) domain (STFT spectrograms) (Lu et al., 2023); see Appendix D.1. Note that FSN+, FRCRN and MP-SENet do not attempt to dereverberate: they infer $r * y$ instead of $y$. For exact checkpoints and detailed architectural notes, see Appendix E.

### 3.2 ADVERSARIAL PERTURBATIONS

Given the definition of speech denoising given in §3.1, an adversarial perturbation for some input $x = r * (y + b)$ is any $\delta \in \mathbb{R}^n$ such that $x + \delta \mapsto y'$, where (a) $y'$ is undesirable and (b) $x + \delta$ sounds identical to $x$. To make these notions concrete, we distinguish untargeted and targeted attacks.

**Untargeted attacks.** Here, we assume that the attacker has a loss function $\mathcal{L} : \mathbb{R}^n \times \mathbb{R}^n \to \mathbb{R}$, and they wish to *maximize* the value of $\mathcal{L}(y', y)$. They do so by selecting $\delta$ from a set of perturbations which are imperceptible when added to $x$, a feasible set which we refer to as $D(x)$. Thus, the adversary in an untargeted attack wishes to find $\delta^* \in \arg\max_{\delta \in D(x)} \mathcal{L}(f(x + \delta), y)$.

**Targeted attacks.** In this case, the attacker has a target output $y'$ that they wish to cause $f$ to output. We model this as the attacker wishing to *minimize* the loss $\mathcal{L}(f(x + \delta), y')$, i.e., the adversary wants $\delta^* \in \arg\min_{\delta \in D(x)} \mathcal{L}(f(x + \delta), y')$.

**Over-the-air attacks.** In an over-the-air attack, the adversarial perturbation is distorted by the room's acoustic characteristics and received through a microphone. As these distortions are assumed to be applied equally to all audio, this is generally simulated by applying the given RIR to the perturbation $\delta$ as well as speech $y$ and background noise $b$ (Qin et al., 2019; Schönherr et al., 2020). For example, the untargeted attack becomes the problem of finding $\delta^* \in \arg\max_{\delta \in D^*(x)} \mathcal{L}(f(r * (y + b + \delta)), y)$, where $D^*$ is all perturbations that are imperceptible after *all* audio, including the perturbation, is convolved with $r$.

### 3.3 AUDITORY MASKING

We use a perceptibility constraint based on auditory masking, also known as psychoacoustic hiding (Lin & Abdulla, 2015; Schönherr et al., 2018; Qin et al., 2019): by varying the loudness constraint on the perturbation according to how loud the original signal is across different time segments and frequency bands, the perturbation can be effectively hidden in the original signal. This is accomplished through the computation of masking thresholds $\theta_{\tau,\omega}$ on the power spectral density (PSD) matrix of the perturbation, such that any audio whose PSD is upper-bounded by these masking thresholds is, in principle, imperceptible by humans. Expanding on the auditory masking algorithms used in this prior work, we apply several enhancements to the threshold calculation to maximize attack power and minimize perceptibility; see §4.2.

## 4 METHOD

Most of the perturbations we study can be characterized as instantiations of the attacks shown in §3.2, given varying settings of $f$, $y$, $b$, $r$, $\mathcal{L}$, $D$, and $y'$. Here, we discuss $\mathcal{L}$, $D$, and optimization.

### 4.1 OPTIMIZATION OBJECTIVE

In this work, we used Short-Time Objective Intelligibility (STOI) (Taal et al., 2011) as the loss function $\mathcal{L}$ for both targeted and untargeted attacks. Concretely, the untargeted objective is

$$\mathcal{L}_{\text{untargeted}}(\delta) = -\text{STOI}(f(x + \delta), y),$$

which we maximize over $\delta \in D(x)$ (see §3.2), driving the model output away from the clean speech $y$. STOI is open-source, differentiable, and closely aligned with our perceptions of intelligibility in untargeted experiments.

While mean-squared error (MSE) is widely used in comparable image tasks and has been successfully employed in prior attacks on generative audio models (Takahashi et al., 2021; Dong et al., 2023), it remains an unreliable metric for intelligibility. It suffers from a lack of several detailed desiderata; for example, it is strongly phase-dependent, meaning that simply delaying the audio by a few milliseconds can cause very large MSE but almost no loss in intelligibility. Nevertheless, the central issue with MSE is that it is simply not designed to be an intelligibility metric; in our experiments, attacks would often achieve a large (untargeted) MSE by simply inducing the DNS model to fail to remove noise, rather than rendering the speech unintelligible.

We could not use Perceptual Evaluation of Speech Quality (PESQ) (Rix et al., 2001), a notable intelligibility metric, as its license owners did not grant us an academic license for this project. However, real adversaries would of course face no such ethical or legal concerns.

Finally, we also considered DNN-based intelligibility metrics such as DNSMOS (Reddy et al., 2022), NISQA (Mittag et al., 2021), and the word error rate (WER) of Whisper (Radford et al., 2022). While such metrics were reasonably effective as unoptimized metrics for attack success,

we found empirically that they work poorly as objectives, as the attack simply learns to induce the model to output adversarial noise against the metric network.

## 4.2 PERCEPTIBILITY CONSTRAINT ENHANCEMENTS

Our precise procedure for computing the masking thresholds is almost identical to the MP3 psychoacoustic model as described by Lin & Abdulla (2015) and used by Schönherr et al. (2018) and Qin et al. (2019), with the following changes: **(a)** for simplicity, we calculate the PSD of the input audio by normalizing to $[-1, 1]$ and converting to dB SPL explicitly, rather than setting the maximum bin to equal 96; **(b)** we enhance the psychoacoustic model with temporal pre- and post-masking, as described by Lin & Abdulla (2015) (see Appendix A for details); **(c)** because prior studies have found even attacks generated according to these masking thresholds to still be detectable by humans (Schönherr et al., 2018; Qin et al., 2019), we restrict our attacks further by shifting all masking thresholds down by 12 dB. We combine thresholds from contemporaneous masking and pre- and post-masking by taking the maximum between each. See Appendices A and B for details.

## 4.3 OPTIMIZATION ALGORITHM

We use projected gradient descent (PGD; Madry et al. 2018) to find the perturbation $\delta^*$:

$$\delta_{t+1} = \Pi_{D(x)} \left( \delta_t + \alpha \frac{\partial \mathcal{L}(f(x + \delta_t), y)}{\partial \delta_t} \right), \tag{1}$$

where $\Pi$ is the projection operator. Fortunately, the projection step is straightforward when using auditory masking on unreverberated perturbations (see §4.4 for details on reverberated perturbation optimization). If $\tilde{\delta}_{\tau,\omega}$ is the STFT spectrogram of $\delta$ at frame $\tau$ and frequency bin $\omega$, then we clip the magnitude of $\tilde{\delta}_{\tau,\omega}$ such that $\text{PSD}(\delta)_{\tau,\omega} \leq \theta_{\tau,\omega}$ while preserving its phase. See Appendix B for details. We also tested Lagrangian dual descent, a variant of which was used by Qin et al. (2019), and the fast gradient sign method (Goodfellow et al., 2015), but these did not perform as well.

## 4.4 ADDITIONAL EXPERIMENTAL METHODS

**Optimization in Simulated Over-the-air Attacks.** We simulate over-the-air attacks in a specific acoustic environment by applying the given RIR to the adversarial perturbation as well as to the clean speech and background noise (see §3.2). Doing so dramatically increases the difficulty of the optimization problem, as the projection step in (1) no longer has a closed-form solution: rather than directly clipping $\delta$, we must instead solve for some $\delta$ such that $\text{PSD}(r * \delta)_{\tau,\omega} \leq \theta_{\tau,\omega}$.

We explore three methods to find such a $\delta$: Wiener deconvolution, manual projection through gradient descent, and a combination of both. See Appendix C for details.

**Targeted Attacks.** We test three types of targets for targeted attacks: speech samples from the same speaker, speech samples from different speakers, and artificial targets generated using voice cloning systems such as MaskGCT (Wang et al., 2024a). For all target types, our attack objective is

$$\mathcal{L}_{\text{targeted}}(\delta) = \text{STOI}(f(x + \delta), y') - \text{STOI}(f(x + \delta), y),$$

which we maximize over $\delta \in D(x)$, so that the model's output becomes more intelligible as the target $y'$ than as the original clean speech $y$.

**Defenses.** Prior work has investigated a wide range of defenses against adversarial audio perturbations, from pre-processing steps like randomized smoothing (Subramanian et al., 2019; Olivier & Raj, 2021), purification (Wu et al., 2023), and adversarial example detection (Yang et al., 2019; Hussain et al., 2021), to more involved defenses like adversarial training (Sallo et al., 2021). In this paper, we evaluate the simplest of these defenses: Gaussian noise applied to the attacked audio, normalized to various SNRs.

**Universal Adversarial Perturbations.** While most of our experiments focus on attacking a single known input, we also explore the existence of universal adversarial perturbations (UAPs): individual perturbations that lead one or more models to misbehave on multiple inputs (Moosavi-Dezfooli et al., 2017; Abdoli et al., 2019; Neekhara et al., 2019; Xie et al., 2021; Zhang et al., 2021; Sun et al., 2024). We design UAPs for DNS models under the assumption that the background noise $b$ and RIR

$r$ remain the same for varying clean speech samples $y_n$; this represents a best case for the attacker, since the perturbation can hide in shared background noise and the masking thresholds need not accommodate dramatically different acoustic scenes. We calculate $D_U = \bigcap_n D(r * (y_n + b))$ (i.e., take the minimum of the different masking thresholds). To train the attack, we iteratively execute one step of PGD sequentially on each input $r * (y_n + b)$.

## 5 EXPERIMENTS

In our computational experiments, we focused on answering the following questions: **a)** Are DNS models susceptible to adversarial noise, or do they denoise that successfully as well? **b)** In which use cases (amount of noise, reverb, etc.) are the models we study vulnerable to adversarial attacks? **c)** How does adversarial robustness vary by model, and which model characteristics seem to correlate with robustness? **d)** Can attackers induce the model to output a target utterance, as in the case of ASR transcriptions, or are they limited to untargeted attacks? **e)** Do imperceptible universal adversarial perturbations exist, or does the attacker need to know what the speaker will say? **f)** Do attackers need to have access to model gradients in order to successfully attack these models? **g)** Are sophisticated defenses (e.g., adversarial training) necessary, or is Gaussian perturbation good enough? **h)** Are real, over-the-air attacks possible in principle, or is this setting too difficult? **i)** How does our perceptibility constraint trade off against attack success as we tighten it (Appendix D.3)?

### 5.1 EXPERIMENTAL DETAILS

**Datasets.** All audio samples—speech, noise, and RIRs—were taken from the main track of the ICASSP 2022 DNS Challenge 4 (Dubey et al., 2022). Ten-second clean speech samples were selected randomly from English read speech (as noted by Dubey et al. (2022), collated from LibriVox.org) and VCTK Corpus (Yamagishi et al., 2019). Audio was clipped to five seconds for MP-SENet due to insufficient VRAM. We filtered speech to contain at least 15 words according to Whisper. See Appendix E for more details.

**Metrics.** In addition to STOI, we evaluated attacked audio and the model outputs using ViSQOL (Hines et al., 2015), an intrusive (binary with ground-truth) and unlearned metric; two non-intrusive deep metrics, NISQA (Mittag et al., 2021) and DNSMOS (Reddy et al., 2022); and word accuracy (de Oliveira et al., 2023) using the ASR model Whisper (Radford et al., 2022). See Figure 2.

**Attacks.** All combinations of environment setting (background SNR and reverb/no reverb) and model choice were run for 20 shared seeds (each corresponding to a distinct speech utterance). The attack lasted for a different number of iterations for each model, chosen to ensure that the entire attack (including metric computation) lasted for about one hour on an Nvidia L40S GPU; in particular, this allowed 20,000 iterations for Demucs and FSN+, 10,000 iterations for MP-SENet, and 5,000 iterations for FRCRN. We standardized compute time rather than iteration count because we view attackability as inclusive of the time cost of mounting an attack: granting a slower model proportionally more optimization steps would obscure the practical advantage that computational expense confers. We verify in Appendix D.6 that fixing iterations preserves robustness rankings.

### 5.2 RESULTS AND DISCUSSION

**Summary of findings.** We briefly answer each experimental question before elaborating in the sections that follow. **a) Susceptibility.** Yes: all four models can be driven to output unintelligible audio by imperceptible perturbations, though FSN+ is partially protected by exploding gradients (Figure 1). **b) Settings.** Attacks succeed across virtually all SNR and reverb settings, including near-clean conditions with 70 dB SNR and no reverb. **c) Robustness.** FSN+ is the most resilient, but due to gradient instability rather than architecture or model size (Appendix D.1); the other three models are comparably susceptible. **d) Targeted.** Mixed: objective metrics suggest success, but subjective listening reveals the target utterance is barely audible in the output (Appendix D.2). **e) UAPs.** No: imperceptible UAPs produce only minor degradation (Appendix D.2). **f) Transfer.** Currently yes: naive cross-architecture transfer fails (Table 2 in Appendix D.2.2), and leave-one-out transfer across Demucs checkpoints is far weaker than white-box attack (Figure 3). **g) Defenses.** Gaussian noise offers partial protection, but only at SNRs that also degrade normal performance, and an adaptive attacker would likely circumvent it (Figure 4). **h) OTA.** Yes: simulated OTA attacks succeed for all

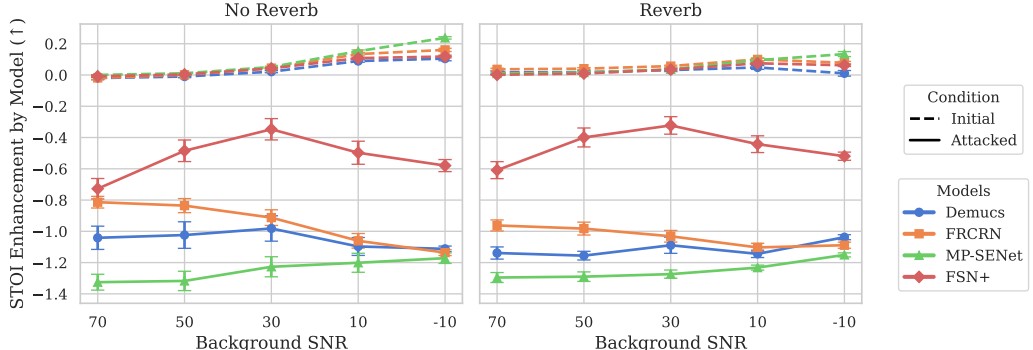

Figure 1: **Untargeted intelligibility degradation.** Dashed curves show the DNS models' *baseline* STOI improvement $\Delta_{\text{STOI}} = \text{STOI}(\text{clean}, \text{output}) - \text{STOI}(\text{clean}, \text{input})$; solid curves show the same quantity after adding an imperceptible perturbation. Values flip from $> 0$ to $< 0$ upon addition of adversarial noise across environmental settings, meaning the attack pushes speech from "cleaner than input" to "less intelligible than the noisy input." Error bars: $\pm$ standard error over 20 seeds.

models except FSN+ (Figure 5), confirmed with real recorded RIRs (Figure 14 in Appendix D.5). **i) Tradeoff.** Tighter masking reduces attack success primarily in low-noise settings; all constraint levels still allow substantial degradation on average (Appendix D.3).

Figure 1 demonstrates our fundamental result: all DNS models we tested could be induced to output far worse-quality audio than the input through the addition of imperceptible adversarial noise, and indeed, all models could be reliably (or often, in the case of FSN+) reduced to outputting gibberish. Furthermore, the success of the attack was relatively invariant with respect to setting: all models could be successfully attacked in all settings, including a setting which has almost no attack vector: $70\,\text{dB}$ SNR noise with no reverb. Thus, we can answer question **b)**: three out of four tested models were completely susceptible in all settings, while FSN+ was fairly susceptible in most settings.

### 5.2.1 MODEL RESULTS

Figures 1 and 7 (Appendix D.2.1) suggest that FSN+ is by far the most resilient, FRCRN and Demucs have comparable resilience, and MP-SENet is slightly more susceptible. Surprisingly, though, the resilience of FSN+ is not due to high-level architecture differences such as domain or parameter count (see Appendix D.1). Instead, the large difference between FSN+'s susceptibility and others was due to the pseudo-robustness of exploding gradients (Athalye et al., 2018): during the course of the attack on FSN+, the gradient of the STOI loss with respect to the adversarial waveform would often grow to have a norm of $10^{30}$ or greater, causing numerical instability even with gradient clipping. Therefore, FSN+ might be more vulnerable to black-box attacks (Athalye et al., 2018).

### 5.2.2 SIMULATED OVER-THE-AIR ATTACKS

While the pseudo-protection of FSN+'s exploding gradients was amplified in our simulated over-the-air experiments, we discovered that all other models are highly vulnerable to untargeted reverberated perturbations, despite the additional optimization challenges in developing them (see Figure 5). However, the increased challenge of this attack did require that we slightly loosen our perceptibility constraint, reducing the masking thresholds by only $6\,\text{dB}$ rather than $12\,\text{dB}$. While this loosening still implies a tighter constraint than prior work, it did cause slight crackling to be audible in the speech, though it would be difficult to distinguish from minor distortion in real settings.

### 5.2.3 DEFENSES

We discovered that simple Gaussian perturbation offers reasonable protection against adversarial perturbations, though only when applied at a low enough SNR to damage model performance (Figure 4; note that a STOI enhancement of 0 is slightly lower than the unattacked average of $\sim 0.044$).

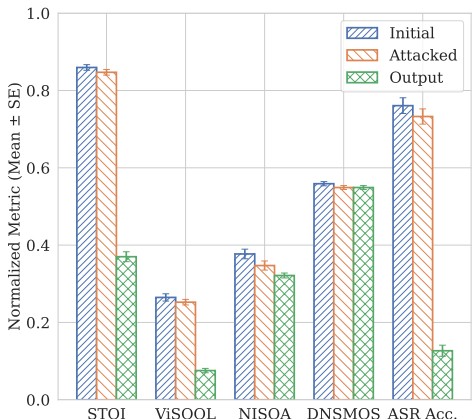

Figure 2: Normalized values of **the five speech intelligibility and quality metrics we test**, averaged across all models and settings of Figure 1 and 20 seeds, with standard error bars. "Output" refers to model output given the attacked input. ASR accuracy is computed as $1 - \min(\text{WER}, 1)$. Ranges used for normalization were: STOI: $[-1, 1]$. ViSQOL: $[1, 5]$. NISQA: $[0, 5]$. DNSMOS: $[0, 5]$. ASR accuracy: $[0, 1]$. ASR ground-truth is Whisper applied to clean speech.

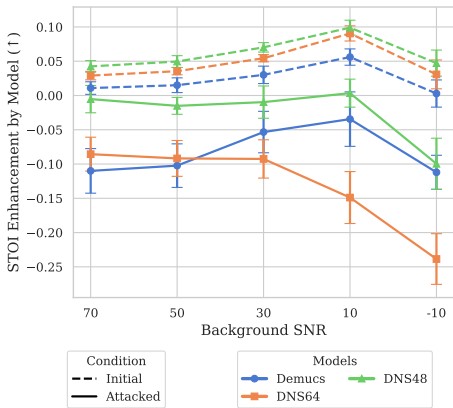

Figure 3: **Leave-one-out transfer across Demucs checkpoints** (`dns48`, `dns64`, and `master64`, labeled 'Demucs') in reverb conditions, using the masking threshold of Qin et al. (2019). Transfer attacks produce minor degradation that grows with background noise, but remain far weaker than white-box attacks (cf. Figure 1). See Figure 8 for full results including no-reverb conditions. Curves show $\Delta_{\text{STOI}} = \text{STOI}(\text{clean}, \text{output}) - \text{STOI}(\text{clean}, \text{input})$; error bars: $\pm$ standard error over 10 seeds.

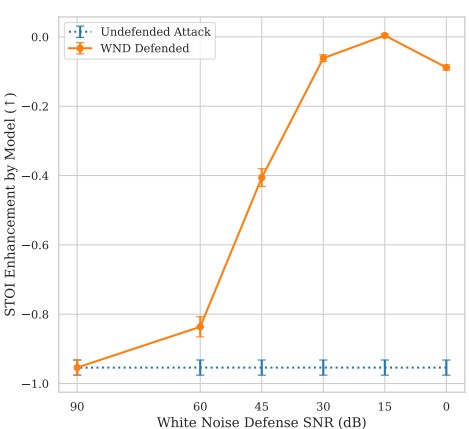

Figure 4: **White-noise defense.** Adding moderate Gaussian ("white") noise to the attacked input partially smooths out adversarial perturbations, raising STOI but not restoring clean performance. Curves show $\Delta_{\text{STOI}} = \text{STOI}(\text{clean}, \text{output}) - \text{STOI}(\text{clean}, \text{input})$; error bars: $\pm$ standard error over 20 seeds.

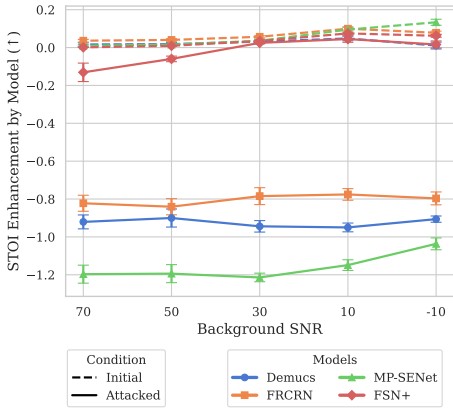

Figure 5: **Simulated over-the-air attack.** Perturbations trained and evaluated under a mix of simulated and real recorded RIRs (OpenSLR26/28) still cripple every model except FSN+. See Figure 14 for results with real RIRs only. Curves show $\Delta_{\text{STOI}} = \text{STOI}(\text{clean}, \text{output}) - \text{STOI}(\text{clean}, \text{input})$; error bars: $\pm$ standard error over 20 seeds.

However, we do not assume adaptation by the attacker; knowledge of the defense might allow them to train more noise-resistant perturbations, emphasizing a need for more sophisticated defenses.

### 5.2.4 ABLATION: ATTACK PIPELINE COMPONENTS

A natural question is which components of our attack pipeline are necessary for imperceptible, successful attacks. To isolate this, we compare five constraint strategies in a single representative setting (30 dB SNR with reverb): (1–2) naive PGD with $\ell_\infty$ and $\ell_2$ bounds in the STFT domain, (3) frequency masking without temporal masking at a calibrated offset, (4) the original masking thresholds of Qin et al. (2019) (frequency masking only, no offset), and (5) our full method (§4.2). For methods 1–3, we calibrate the constraint so that the same ΔSTOI is achieved as our full method, isolating the perceptibility cost: simpler methods must allow louder, more audible perturbations to reach the same attack success. We provide details and links to samples in Appendix D.4.

## 6 HUMAN STUDY

**Participants & ethics.** We recruited 15 adult audio/multimedia researchers from an industrial research lab. Participation was voluntary and conducted during paid working hours; no additional compensation was provided. Under the host institution's policy for minimal-risk studies, our study did not require formal IRB/ethics review. All participants provided informed consent and could withdraw at any time, and no personally identifying information or audio recordings were collected; we stored only anonymized task responses and timestamps.

**Tasks and protocol.** Each participant completed two tasks in a fixed order: (1) *transcription* and (2) *ABX discrimination*. Participants could take as long as they wished, though tasks were designed to be completed in less than 25 minutes. Once a participant advanced from transcription to ABX, they could not return to transcription. Participants typically use high-quality headphones in their day-to-day work; we recommended (but did not enforce) headphone use. Participants were instructed to leave transcription boxes blank if no intelligible speech was detected.

**Stimuli and conditions.** All stimuli were 5 s clips cropped from the same datasets as in the computational study (pilot feedback indicated 10 s was too long to transcribe or compare reliably). We filtered candidate clips to contain at least 10 words and to use only the 2,000 most common English words to reduce ESL burden, again thanks to pilot feedback. We used two background conditions designed to mimic common listening scenarios: 30 dB and 50 dB SNR with reverberation. We evaluated Demucs, FRCRN, and MP-SENet in the human study; Full-SubNet+ was excluded here so as to focus our human-evaluation budget on the most vulnerable models.

**Attacks.** Attacks were conducted as in the computational study, here using the `tighter` psychoacoustic masking constraint (Appendix D.3).

**Measures and analysis.** For transcription we report word accuracy (WAcc; $1 - $ WER), calculated without any equivalencies besides uppercase–lowercase. For ABX we report accuracy versus the 50% chance baseline. For confidence intervals, we use 95% two-way ('pigeonhole') bootstrap CIs (Owen, 2007), a model-free alternative to generalized linear mixed models (GLMMs) for data with crossed effects. Note that, because samples were shared between participants, naive bootstrap and the Student's $t$-test would not be appropriate statistical tests, as they would underestimate variance.

**Results.** Transcription WAcc (Figure 6a) shows that the **Attacked Output** had near-zero intelligibility, while both the **Attacked Input** and **Clean Output** were reasonably intelligible. In particular, an intersection-union test indicates that the attacked output was significantly less intelligible than both the attacked input and clean output (95% upper bounds on the mean differences of $-0.464$ and $-0.458$, respectively; see Appendix F for details).

ABX accuracy (Figure 6b) had a mean of 59%, marginally above chance, but with substantial variance between participants and samples. The mean was not significantly different from 50% according to a one-sided 95% pigeonhole bootstrap test (lower bound: 0.478). While other less conservative statistical tests (e.g., a Wald test with a GLMM) might conclude significance, the present results offer preliminary support for the subjective imperceptibility of our attacks.

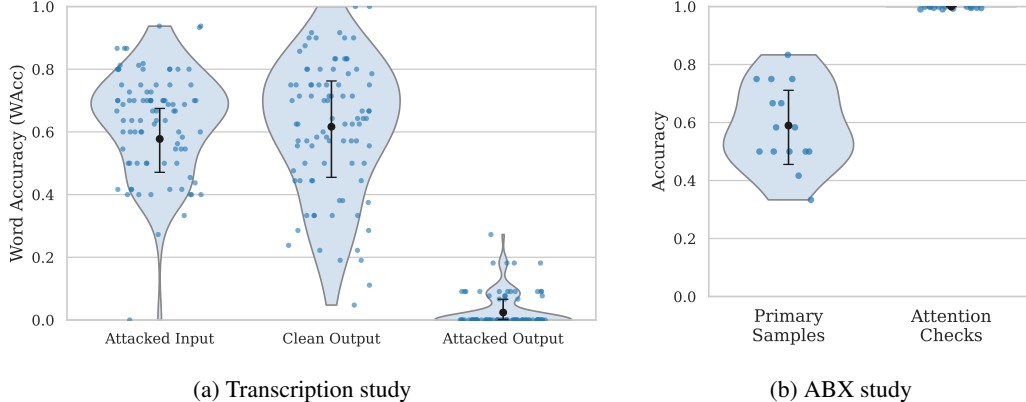

(a) Transcription study        (b) ABX study

Figure 6: **Human study**. We ran transcription and ABX studies with 15 audio/multimedia researchers using high-quality audio setups. The transcription task consisted of 18 random 5-second samples (sampled once for all participants), including 6 attacked inputs, 6 model outputs given clean inputs, and 6 model outputs given attacked inputs. Word accuracy (WAcc) results show that the attacked input and clean output were reasonably intelligible, while the attacked output was unintelligible. The ABX task consisted of 12 pairs of 5-second samples (also common across participants), each consisting of an unattacked sample and its attacked counterpart, as well as two attention checks each consisting of a pair of clean and aggressively attacked audio. For both plots, we show 95% two-way bootstrap CIs (Owen, 2007); for the ABX study, we plot participant-wise marginal accuracies (i.e., averaged across samples). ABX results show that the average participant's accuracy is insignificantly above the random chance baseline of 50%, with substantial variance.

## 7    CONCLUSION

We show that DNS models are vulnerable to imperceptible adversarial attacks across simulated settings: by adding adversarial noise hidden in the original input signal, all models we study can be induced to output audio with almost no resemblance to clean or noisy speech. Our attacks generalize across settings, including simulated over-the-air attacks and cases with no background noise or reverb, and suggest that better optimization targets may enable subjectively effective targeted attacks (Appendix D.2). Finally, our human study validates that our attacks are generally imperceptible, and the outputs generally unintelligible, even to audio experts.

Our results have several important limitations. First, FSN+ is protected from gradient-based attacks due to exploding gradients, though this pseudo-defense is easily overcome (Athalye et al., 2018). Second, our strongest attacks rely on gradient access; naive transfer is ineffective under strict masking. Third, we only attack fully differentiable DNS models; token-based DNS models (Wang et al., 2024b) will require new techniques. More broadly, our attacks are offline and per-utterance: a streaming attacker would need universal perturbations (ineffective here), a lookahead buffer, or advance knowledge of the speech to attack. Our over-the-air experiments model linear propagation via RIRs (real recorded RIRs subsume speaker/microphone frequency responses), but real playback chains also introduce nonlinearities, gain control, and codec compression (Sadasivan et al., 2025), any of which could attenuate the perturbation. Weak transfer suggests a defense: inference-time ensembling or randomized architecture switching could raise the bar for an attacker, even one with white-box access to each individual model.

Nevertheless, we hope to convince the research community that open-source DNS models are both an appealing and feasible target for attack, with the potential to incapacitate live audio streams, speech recognition systems, hearing aid users, and more. With simple defenses such as white noise only offering limited protection, we urge researchers to evaluate further models in more lifelike settings and develop and apply better defenses before attackers exploit these critical weaknesses.

## ACKNOWLEDGEMENTS

We are deeply grateful to Neel Chaudhari (Dolby Laboratories) for his invaluable work designing, implementing, and deploying the perceptual listening-test platform used in our human-subjects evaluation.

Funding for this project was generously provided by Dolby Laboratories. This project also used computational resources from Unity, a collaborative, multi-institutional high-performance computing cluster managed by UMass Amherst Research Computing and Data.

## ETHICS STATEMENT

**Dual-use adversarial robustness research.** This paper demonstrates research into the adversarial robustness of DNS models, including methodological advancements in adversarial perturbation generation. However, we believe the security utility of our present work outweighs any utility it could provide to real-world adversaries. First, the focus of the study is on precisely characterizing long-standing vulnerabilities so as to spur future research rather than providing methods for overcoming existing defenses. Second, we indeed document that there are several further steps required for an attacker before successfully applying the present methods to a real-world scenario, including extending the simulated over-the-air attacks shown here to real over-the-air scenarios. Finally, we show that simple, easily implemented defenses such as Gaussian perturbation are effective against naive attacks, thus leaving the immediate advantage with a well-informed defender.

**Human subjects.** The present paper also describes a small human study with 15 industry researchers. Participation was purely voluntary during business hours, with compensation only in the form of their existing salary; explicit consent was collected before each participant began the study; the study consisted purely of audio tasks containing no offensive speech; each participant was free to withdraw at any time without penalty; and no personal information was collected or stored at any time. While the host institution of the study does not require IRB approval for low-risk studies, we believe the risks to subjects posed by our study design are minimal.

**Licensing and credit**. We carefully track the licenses of all artifacts we use other than standard PyPI libraries (Appendix G). In particular, we did not use the proprietary speech quality metric PESQ, as it is not licensed for research use by default and the maintainers did not respond to our request.

## LLM DISCLOSURE

LLMs provided assistance throughout this project except for ideation of the central research questions. LLMs from the GPT, Claude, and Gemini families were used for methodological ideation, writing, and coding assistance. The authors attest to the correctness of all present claims and results.

## REPRODUCIBILITY STATEMENT

We have specified all details needed to reproduce our experiments to the best of our ability both in the main body and in all appendices. Our code is publicly available at https://github.com/willschwarzer/adv-dns-public.

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

## A  AUDITORY MASKING

**Temporal Masking.** Let $\theta_{\tau,\omega}$ be the masking threshold in frame $\tau$ and frequency bin $\omega$, as computed by the default approach (Lin & Abdulla, 2015; Qin et al., 2019). This computation already involves consideration of the effect of sound in one frame $\tau$ and frequency bin $\omega$ on the masking thresholds in the same *frame*, $\tau$, but adjacent *frequency bins*, $\omega'$. For example, the model already accounts for the fact that a 1 kHz sine tone would mask a 990 Hz tone as well.

However, it is well known (Lin & Abdulla, 2015) that masking also occurs between *frames*, along the same *frequency bin*; this is known as pre-masking and post-masking. Combining results shown by Lin & Abdulla (2015) with our own experimentation, we model this by first computing contemporaneous masking thresholds as before, then for each $\theta_{\tau,\omega}$ computing the post-masking threshold it causes on later frames $\tau' > \tau$ using exponential decay of $0.02\,\mathrm{ms}^{-1}$, clipping to zero at a gap of $100\,\mathrm{ms}$. Pre-masking, where $\tau' < \tau$, is modeled similarly but much more sharply, with a decay of $0.16\,\mathrm{ms}^{-1}$, clipping to zero after a gap of $20\,\mathrm{ms}$.

Finally, each frame and frequency bin now has one contemporaneous masking threshold, a small number of masking thresholds due to pre-masking by future sounds, and a larger number of masking thresholds due to post-masking by previous sounds. To combine all of these thresholds, we set the final threshold to be their maximum, as we find this to empirically produce the best perceptibility constraint.

## B  PROJECTED GRADIENT DESCENT WITH MASKING

Given the TF masking thresholds $\theta_{\tau,\omega}$ calculated for the normalized input $x$, we seek an imperceptible perturbation $\delta$ whose reverberation-free STFT magnitude does not exceed $\theta_{\tau,\omega}$. Let

$$\widetilde{\delta}_{\tau,\omega} = \mathrm{STFT}(\delta)_{\tau,\omega}.$$

For the purposes of illustration, let

$$\mathrm{PSD}(\delta)_{\tau,\omega} = 10\log_{10}\big(|\widetilde{\delta}_{\tau,\omega}|^2\big).$$

(In reality, there are subtleties around Hann window energy and length correction which we elide here; the full process is discussed by Lin & Abdulla (2015) and Qin et al. (2019).)

To enforce $\mathrm{PSD}(\delta)_{\tau,\omega} \leq \theta_{\tau,\omega}$, we use a projection operator $\Pi_{D(x)}$ in each iteration of projected gradient descent (PGD). Specifically, after each gradient step, we adjust $\widetilde{\delta}_{\tau,\omega}$ to satisfy

$$\mathrm{PSD}(\delta)_{\tau,\omega} \leq \theta_{\tau,\omega}$$

by scaling its magnitude if necessary:

$$\widetilde{\delta}_{\tau,\omega} \leftarrow \widetilde{\delta}_{\tau,\omega} \times \min\Big(1, 10^{\frac{\theta_{\tau,\omega}-\mathrm{PSD}(\delta)_{\tau,\omega}}{20}}\Big).$$

This enforces time-frequency masking constraints bin-by-bin while preserving the phase of $\widetilde{\delta}_{\tau,\omega}$.

Note that this is not necessarily a projection operator in the strict sense of mapping to the nearest point in $D(x)$, particularly when $\delta$ is parameterized in the time domain, but it does ensure a feasible $\delta$ after each projection step.

## C  OVER-THE-AIR ATTACK OPTIMIZATION

Given knowledge of an RIR $r$ and a clipped, imperceptible reverberated perturbation $\delta^r = r * \delta$, Wiener deconvolution is the following process: if $\tilde{\delta}^r_\omega$ is the full frequency spectrogram of $\delta^r$ in bin $\omega$, and similar for $\tilde{r}_\omega$, we approximate

$$\tilde{\delta}_\omega \approx \frac{\tilde{\delta}^r_\omega \tilde{r}^*_\omega}{|\tilde{r}^*_\omega|^2 + \epsilon},$$

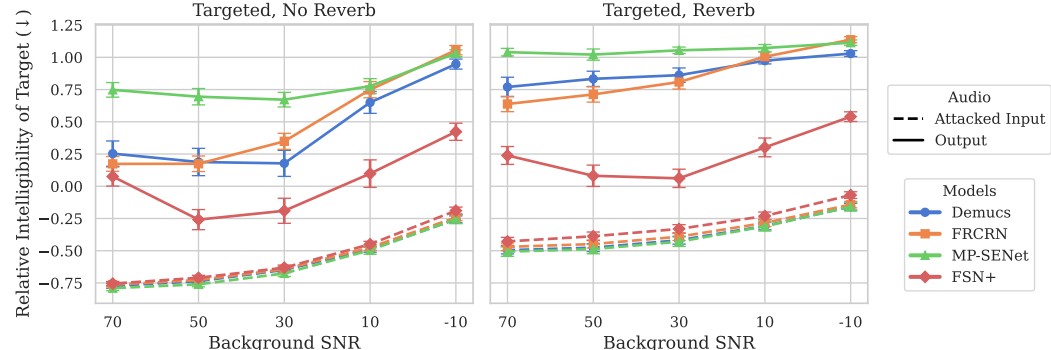

Figure 7: **Intelligibility of injected target audio.** We plot $\Delta_{\text{target}} = \text{STOI}(\text{target}, \text{audio}) - \text{STOI}(\text{clean}, \text{audio})$ for the attacked input (dashed) and the DNS output (solid). Positive solid values indicate that the model's output makes a *hidden* target phrase clearer than the original clean speech, demonstrating a successful targeted attack under the STOI criterion. Targets are real utterances from the same speaker as the clean audio. Error bars: $\pm$ standard error over 20 seeds.

where $\epsilon$ is a small stability term ($10^{-4}$) and $*$ represents complex conjugation. Although a mathematically principled inversion, Wiener deconvolution is approximate for our use case, as convolution by RIRs is generally non-invertible.

One could attempt to apply Wiener deconvolution iteratively to find an exact $\delta$ such that $(r * \delta) \in D^*(x)$. However, a simpler iterative approach is gradient descent: we directly compute the total constraint violation $g(\delta)$ as

$$g(\delta) = \sum_{\tau, \omega} \max(\text{PSD}(r * \delta)_{\tau, \omega} - (\theta_{\tau, \omega} - d), 0),$$

where $d$ is a small positive offset ($1\,\text{dB}$) to ensure that gradient descent finds an exact feasible solution; we then find $\frac{\partial g(\delta)}{\partial \delta}$ using autodifferentiation and repeat until $\delta \in D^*(x)$, i.e., $g(\delta) = 0$. For over-the-air attacks, we parameterize $\delta$ in STFT spectrogram space, as this representation produces smoother gradient descent behavior. Because these attacks are harder, we gradually decrease the masking thresholds from a high starting point over time, similar to iterative constraint tightening in other works (Qin et al., 2019).

Finally, we also test applying one step of Wiener deconvolution and then finishing the projection with gradient descent.

## D    ADDITIONAL EXPERIMENTS

### D.1    MODEL ATTACK SUCCESS VERSUS ARCHITECTURAL DETAILS

Table 1: **Effect of architecture on robustness to adversarial perturbations.** FSN+ shows substantially smaller STOI degradation than other models with similar domains or parameter counts, so coarse architectural statistics do not predict robustness.

| Model | Domain | #Par. (M) | $\Delta$STOI |
|-------|--------|-----------|--------------|
| Demucs | Time | 33.5 | $-1.08$ |
| FRCRN | TF | 10.3 | $-0.99$ |
| FSN+ | TF | 8.7 | $-0.49$ |
| MPSE | TF | 2.3 | $-1.25$ |

### D.2    NEGATIVE RESULTS

In this section, we explore several of our research questions that presented mixed or negative results.

**Universal Adversarial Perturbations (UAPs).** We found that UAPs were unable to cause more than slight decreases in the quality of the model output while remaining imperceptible (question e)); this is likely due to the difficulty of hiding relevant perturbations inside of background noise alone, rather than a single clean speech sample. This result is consistent with prior work showing that audio UAPs are generally perceptible (Sun et al., 2024).

### D.2.1 TARGETED ATTACKS

Figure 7 demonstrates our results for the strongest targeted attack, where targets are real samples from the same speaker. At first glance, it appears to suggest that the answer to question **d)** is resoundingly positive: DNS models are equally susceptible to targeted attacks as ASR models, and equally susceptible to adversarial perturbations in general. However, we found that, while STOI was an excellent minimization objective (low STOI implies low intelligibility), it broke down when used as a metric for maximization (high STOI does not imply high intelligibility). This issue is audible in our samples: even in attacked model outputs that are judged to have STOIs of more than $0.5$ with respect to the target audio, a human listener can discern at most a faint robotic hint of the target speech. Thus, while future attackers may find ways to generate targeted attacks using more sophisticated objectives, the answer to question d) is currently both positive and negative.

Note that cross-speaker targets and AI-synthesized targets were empirically ineffective, so we only report results using real speech from the same speaker.

### D.2.2 MODEL TRANSFER

To answer question f), we evaluated the possibility of model transfer attacks, where an attack is trained for one model but applied to another. Our results indicated that attacks generally do *not* transfer between different architectures (e.g., from Demucs to FSN+); see Table 2. We also tested whether attacks between three different available Demucs checkpoints transfer to each other. Surprisingly, we found that they did not.

Our results stand in contrast to those of Dong et al. (2023), who found that perceptible adversarial examples can transfer between models. Thus, without more sophisticated transfer techniques (Guo et al., 2023), our results suggest an affirmative answer to question f), *as long as* the attacker is truly constrained to producing imperceptible attacks. However, further research is urgently required, particularly on pure black-box attacks (Chen et al., 2021).

Table 2: **Psychoacoustically hidden attacks do not transfer between models.** Off-diagonal entries in the attack-transfer matrix are close to 0 despite similar training data, indicating that gradient access is required for effective imperceptible attacks and that *naive* cross-architecture transfer is weak.

| Trained on | Evaluated on | | | |
|---|---|---|---|---|
| | Demucs | FSN+ | FRCRN | MPSE |
| Demucs | $-1.08$ | 0.04 | 0.06 | 0.08 |
| FSN+ | 0.05 | $-0.49$ | 0.05 | 0.08 |
| FRCRN | 0.06 | 0.04 | $-0.99$ | 0.08 |
| MPSE | 0.03 | 0.03 | 0.05 | $-1.25$ |

To give transfer the best possible chance, we additionally performed a leave-one-out experiment across three Demucs checkpoints (dns48, dns64, and master64), training on two models and evaluating on the held-out model. We also raised the masking threshold back to the original level used in Qin et al. (2019) to further advantage the attacker. As shown in Figure 8, attacks trained on multiple similar models have minor success against held-out models—corresponding to a worsening of speech quality, but not to the point of unintelligibility—and dramatically less success than under white-box gradient access. However, attack success does increase with background noise level and reverb, suggesting that environmental masking partially compensates for the loss of gradient information.

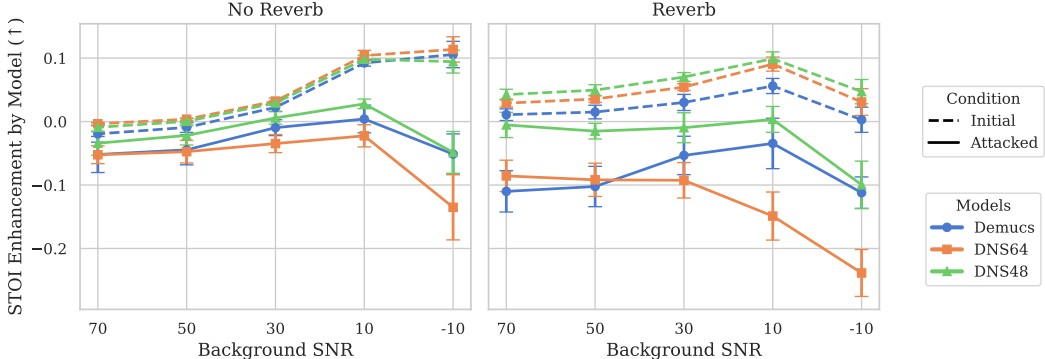

Figure 8: **Leave-one-out transfer across Demucs checkpoints.** Perturbations are optimized on two of the three available Demucs checkpoints (dns48, dns64, master64) and evaluated on the held-out model, using the original masking threshold of Qin et al. (2019). Transfer attacks produce minor degradation that grows with background noise and reverb, but remain far weaker than white-box attacks (cf. Figure 1). Curves show $\Delta_{\mathrm{STOI}} = \mathrm{STOI}(\mathrm{clean}, \mathrm{output}) - \mathrm{STOI}(\mathrm{clean}, \mathrm{input})$; error bars: $\pm$ standard error over 20 seeds.

### D.3 Adjusting perceptibility constraints

**Setup.** In this section, we explore the effect on our objective metrics of adjusting the perceptibility constraint described in the main text. We do this primarily by adjusting the masking offset, setting it to either $-8\,\mathrm{dB}$ (looser), $-12\,\mathrm{dB}$ (default, same as experiments in main text), and $-16\,\mathrm{dB}$ (tighter). For the 'tighter' setting, we also double the decay constant used in post-masking (see Appendix A) to $0.04\,\mathrm{ms}^{-1}$. To simplify presentation, we show results for each constraint setting averaged across the four models. All other experimental details are the same as in the main experiments.

Note that, as in the experiment shown in the main text, simulated OTA attacks use an effectively looser masking constraint by about $6\,\mathrm{dB}$ at the end of optimization (via OTA-specific projection-violation tolerance and best-sample selection), relative to the other experiments. Furthermore, one simulated over-the-air attack in Figure 13 failed to converge in a reasonable amount of time (3 hours on an L40S); we therefore set its final STOI enhancement value to be that of the initial unperturbed input.

**Results.** We include samples from an attack on a moderately noisy, no-reverb speech sample using each of the perceptibility constraints described here at https://sites.google.com/view/adv-dns/perceptibility. Subjective listening evaluation suggests that the looser constraint results in audible crackling in the sample, which goes away completely as the constraints are tightened.

Overall, our objective results suggest that the specific setting of the perceptibility constraint matters much more in relatively low-degradation environments (no reverb and moderate-to-low SNR); see Figures 9 and 10. This is unsurprising, as high-degradation environments naturally offer more space to hide the adversarial perturbation. When averaged across environments, though, the setting of the perceptibility constraint makes a significant difference in success, though all constraints we test allow substantial success (Figure 11), as measured by metrics other than NISQA and DNSMOS. Interestingly, most metrics (except NISQA) suggest that there is no statistically significant difference in audio quality between attacked inputs under the various constraints, implying that even the loosest constraint damages audio only very slightly.

Unsurprisingly, the white noise defense is proportionally roughly equally effective against attacks under all perceptibility constraints (Figure 12): as implied by Figure 11, even the loosest constraint still only allows very slight perturbations, which are therefore largely smoothed out by $15\,\mathrm{dB}$ white noise. ($0\,\mathrm{dB}$ white noise smooths out the signal as well, hence its worse performance.)

Finally, we found that the perceptibility constraint had less effect on simulated over-the-air attack success (Figure 13) than on our directly applied attacks (Figure 9), mostly due to loosening of the

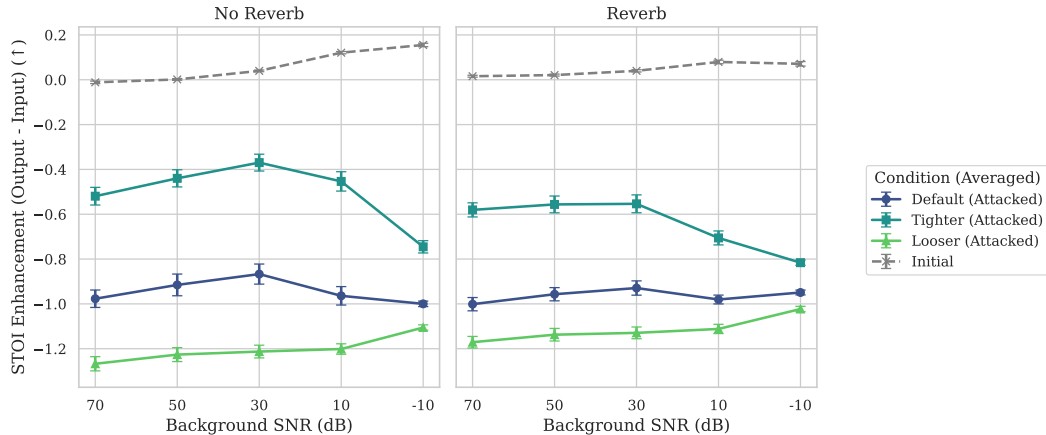

Figure 9: **Untargeted intelligibility degradation across perceptibility constraints.** The dashed curve shows the DNS models' *baseline* STOI improvement $\Delta_{\text{STOI}} = \text{STOI}(\text{clean}, \text{output}) - \text{STOI}(\text{clean}, \text{input})$; solid curves show the same quantity after adding an imperceptible perturbation. Values flip from $>0$ to $<0$ upon addition of adversarial noise across environmental settings, meaning the attack pushes speech from "cleaner than input" to "less intelligible than the noisy input itself." Values are averaged across all four models. Error bars: $\pm$ standard error over 20 seeds.

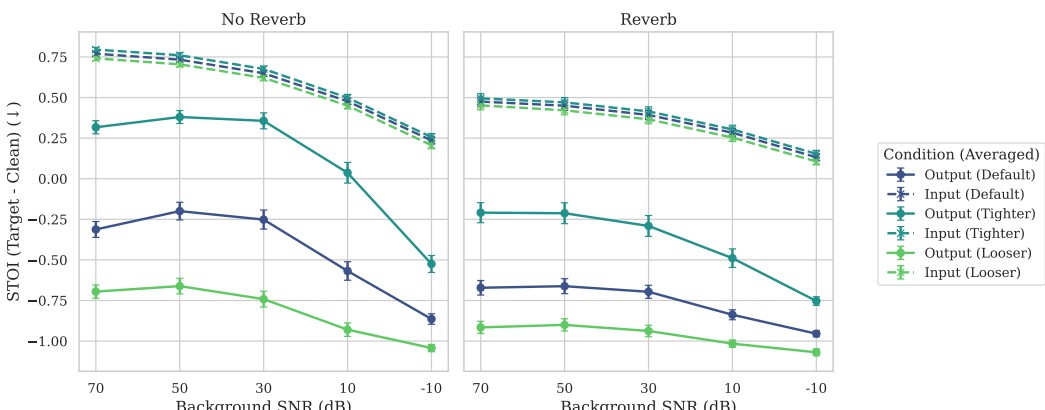

Figure 10: **Intelligibility of injected target audio across perceptibility constraints.** We plot $\Delta_{\text{target}} = \text{STOI}(\text{target}, \text{audio}) - \text{STOI}(\text{clean}, \text{audio})$ for the attacked input (dashed) and the DNS output (solid). Positive solid values indicate that the model's output makes a *hidden* target phrase clearer than the original clean speech, demonstrating a successful targeted attack. Targets are real utterances from the same speaker as the clean audio. Values are averaged across all four models. Error bars: $\pm$ standard error over 20 seeds.

constraint not seeming to allow for significantly stronger attacks. We are not certain why this is the case; while this phenomenon may be specific to the particular OTA attack optimization algorithm we use, it may also indicate more generally that convolution with an RIR—which can indeed be considered a form of smoothing—may represent a defense against adversarial perturbations, including even those which are designed adaptively against that RIR.

### D.4 ABLATION: ATTACK PIPELINE COMPONENTS

To isolate the contribution of each component in our perceptibility constraint pipeline, we compare our full method against several simpler alternatives. We fix the environment to $30\,\text{dB}$ background SNR with reverb, using the Demucs (`master64`) model and the speech sample selected by our first seed, and optimize for 5,000 iterations.

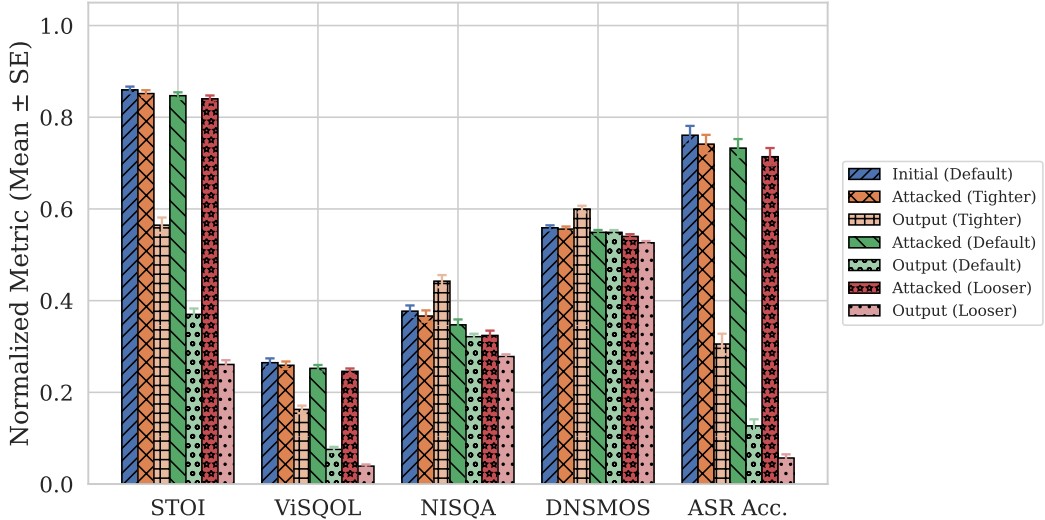

Figure 11: **Additional metrics across perceptibility constraints.** Normalized values of various speech intelligibility and quality metrics, averaged across all models and environmental settings and 20 seeds, with standard error bars. "Output" refers to model output given the attacked input. ASR accuracy is computed as $1 - \min(\mathrm{WER}, 1)$. Ranges used for normalization were: STOI: $[-1, 1]$. ViSQOL: $[1, 5]$. NISQA: $[0, 5]$. DNSMOS: $[0, 5]$. ASR accuracy: $[0, 1]$. ASR ground truth is Whisper applied to clean speech.

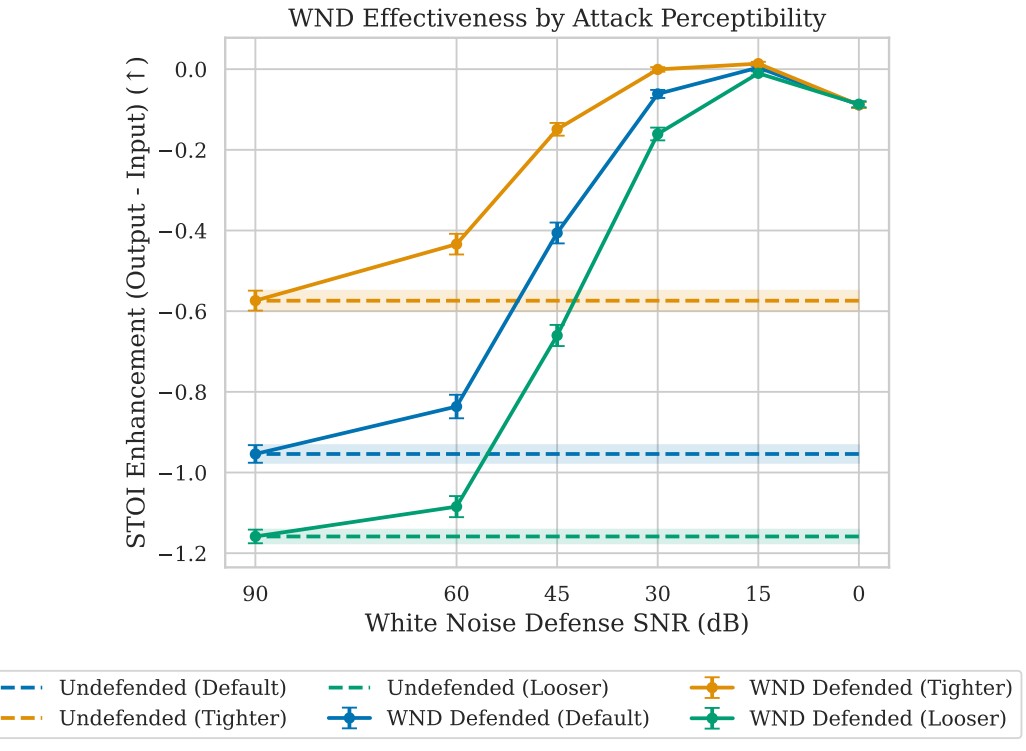

Figure 12: **White noise defense performance by perceptibility constraint.** Dashed lines show the DNS models' attacked STOI improvement $\Delta_{\mathrm{STOI}} = \mathrm{STOI}(\mathrm{clean}, \mathrm{output}) - \mathrm{STOI}(\mathrm{clean}, \mathrm{input})$; solid curves show the same quantity when the attacked audio is subjected to Gaussian perturbation, i.e., addition of white noise of varying signal-to-noise ratios (SNRs).

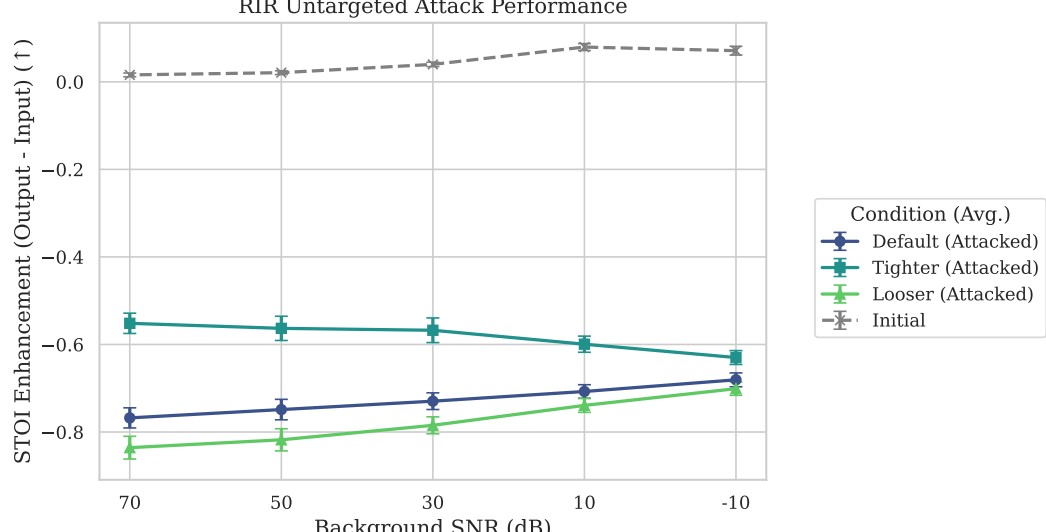

Figure 13: **Simulated over-the-air attack success across perceptibility constraints.** The dashed curve shows the DNS models' *baseline* STOI improvement $\Delta_{\mathrm{STOI}} = \mathrm{STOI}(\mathrm{clean}, \mathrm{output}) - \mathrm{STOI}(\mathrm{clean}, \mathrm{input})$; solid curves show the same quantity after adding an imperceptible perturbation, passed through *the same* room impulse response (RIR) as the original audio itself. Values flip from $> 0$ to $< 0$ upon addition of adversarial noise across environmental settings, meaning the attack pushes speech from "cleaner than input" to "less intelligible than the noisy input itself." Values are averaged across all four models. Error bars: $\pm$ standard error over 20 seeds. Effective OTA masking constraints are about $6\,\mathrm{dB}$ looser than in the other experiments.

**Methods compared.** We compare five constraint strategies:

1. $\ell_\infty$**-norm in STFT domain** ($\varepsilon = 0.04$): PGD with a simple magnitude bound on the STFT of the perturbation, without any psychoacoustic modeling.

2. $\ell_2$**-norm in STFT domain** ($\varepsilon = 18$): PGD with a global $\ell_2$ bound on the STFT magnitudes.

3. **Frequency masking only** (offset $= -8.4\,\mathrm{dB}$): psychoacoustic frequency masking thresholds *without* temporal pre- and post-masking, with the masking offset calibrated to approximately match the attack success of our full method.

4. **Frequency masking, no offset** (offset $= 0\,\mathrm{dB}$): frequency masking without temporal masking and without shifting thresholds below the psychoacoustic model's raw output, corresponding exactly to the constraint used by Qin et al. (2019).

5. **Full method** (offset $= -12\,\mathrm{dB}$, temporal masking): our complete pipeline, including temporal pre- and post-masking (§4.2) and a $12\,\mathrm{dB}$ threshold reduction.

**Calibration procedure.** Our full method (5) achieves $\Delta_{\mathrm{STOI}} \approx -1.1$ at 5,000 iterations. For methods 1–3, we swept over constraint parameters and selected the tightest setting that achieved at least this level of attack success; note that the resulting methods may slightly overshoot. Method 4 uses the original thresholds of Qin et al. (2019) without calibration. All methods use the same PGD optimizer, learning rate schedule, and initialization.

**Results.** All five methods achieve substantial attack success, driving the model's STOI improvement well below zero. The key difference is in *how audible the perturbation is*. For methods 1–2, the $p$-norm constraints bear no relation to human perception, and the resulting perturbations—while small in norm—contain energy at perceptually salient frequencies and time intervals. Method 3 respects frequency masking thresholds but lacks temporal masking; because temporal pre- and post-masking raises the masking threshold near signal transients (where the ear is briefly less sensitive), omitting it reduces the overall perturbation budget. To compensate, the masking offset must be relaxed to $-8.4\,\mathrm{dB}$ (from $-12\,\mathrm{dB}$) to achieve comparable attack success. This $3.6\,\mathrm{dB}$ gap quantifies

the additional perturbation budget that temporal masking provides. Method 4, with no threshold reduction at all, achieves the strongest attack ($\Delta_{\text{STOI}} \approx -1.4$) but produces the most perceptible artifacts. Our full method (5) achieves comparable attack success to the calibrated baselines while keeping the perturbation within the tightest constraint.

We encourage the reader to listen to samples from each method at https://sites.google.com/view/adv-dns/ablation to judge the perceptual differences firsthand.

### D.5 REAL RECORDED RIR ROBUSTNESS

To verify that our simulated over-the-air results are not artifacts of synthetic RIRs, we re-ran our attacks using only real, recorded room impulse responses from the OpenSLR28 dataset (Ko et al., 2017), keeping perceptibility constraints identical to the main experiments. As shown in Figure 14, the attacks remain effective under real RIRs: STOI is still dramatically damaged ($> 0.4$ decrease from input) at background SNRs of $30\,\text{dB}$ or noisier, confirming that the vulnerabilities we identify generalize beyond simulation. Attack success does depend more strongly on the volume of background noise than in the simulated case, likely reflecting the greater acoustic diversity present in real recordings.

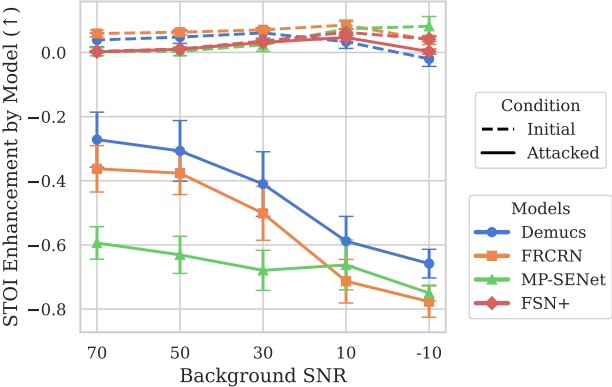

Figure 14: **Over-the-air attack with real recorded RIRs.** Same experimental setup as Figure 5, but using only real recorded RIRs from OpenSLR28 rather than a mixture of simulated and recorded RIRs. Attacks remain effective, though success depends more on background noise level than in the simulated case. Curves show $\Delta_{\text{STOI}} = \text{STOI}(\text{clean}, \text{output}) - \text{STOI}(\text{clean}, \text{input})$; error bars: $\pm$ standard error over 20 seeds.

### D.6 FIXED-ITERATION SANITY CHECK

As noted in Section 5, our main experiments allocate a different number of PGD iterations to each model so that total GPU time is approximately equal. To verify that this choice does not confound the robustness ranking, we rerun the attack with a *fixed* budget of 5,000 iterations for every model, using the same 20 seeds at background SNR $= 30\,\text{dB}$ with reverb. Table 3 reports STOI enhancement ($\Delta_{\text{STOI}}$) before and after the attack.

Table 3: Fixed-iteration sanity check (5,000 iterations per model, SNR $= 30\,\text{dB}$, reverb). Values are mean $\pm$ standard deviation of $\Delta_{\text{STOI}} = \text{STOI}(\text{clean}, \text{output}) - \text{STOI}(\text{clean}, \text{input})$ over 20 seeds. Higher (less negative) attacked enhancement indicates greater robustness.

| Model | Initial $\Delta_{\text{STOI}}$ | Attacked $\Delta_{\text{STOI}}$ |
|---|---|---|
| FSN+ | $+0.035 \pm 0.022$ | $\mathbf{-0.338 \pm 0.245}$ |
| FRCRN | $+0.057 \pm 0.024$ | $-1.034 \pm 0.159$ |
| Demucs | $+0.031 \pm 0.041$ | $-1.082 \pm 0.183$ |
| MP-SENet | $+0.036 \pm 0.040$ | $-1.258 \pm 0.102$ |

The ranking is consistent with the main results: FSN+ remains by far the most robust model, followed by FRCRN, Demucs, and MP-SENet. These results suggest that FRCRN's robustness relative to Demucs and MP-SENet is not due to slower attack generation.

# E    ADDITIONAL EXPERIMENTAL DETAILS

## E.1    MODEL DETAILS

**Demucs.** The earliest of the models we study, Demucs—specifically, the `master64` Demucs checkpoint provided in the Denoiser library—is a time-domain denoising model with 33.5M (trainable) parameters. Uniquely among the models we study, Demucs operates end-to-end on waveforms, rather than spectrograms: it processes the input waveform using several convolutional layers, then encodes temporal dependencies with an LSTM, before decoding again with convolutional layers. Unlike the other models we study, Demucs is designed to dereverberate as well as denoise audio.

**Full-SubNet+.** Full-SubNet+ (FSN+) is a TF-domain denoising model with 8.67M parameters. Full-SubNet+ takes magnitude, real, and imaginary spectra as input, and passes them through a variety of modules, including attention, convolution, and LSTMs. Like all TF-domain models we study, Full-SubNet+ outputs a *ratio mask*, a complex-valued spectrogram $q$ such that $\hat{y}$ equals the element-wise complex product of $x$ and $q$ (Williamson et al., 2016).

**FRCRN.** FRCRN is a TF-domain model with 10.3M parameters. Similar to FSN+, FRCRN uses a mixture of convolutional, attention and recurrent structures in its architecture.

**MP-SENet.** MP-SENet (MPSE)—specifically, the `DNS` checkpoint provided in the official GitHub repository—is a TF-domain model with only 2.26M parameters. Similar to FSN+ and FRCRN, MP-SENet also uses convolution, attention and recurrence for temporal modeling.

Note that, as mentioned in the main paper, MP-SENet used an unusually large amount of memory in our experiments, forcing us to curtail its input audio to 5 seconds. A note was later added to the MP-SENet README explaining that this is due to a bug in the behavior of the model with default arguments, and can be solved with alternative arguments; unfortunately, this note was added after we ran our experiments, so our experiments still curtail MP-SENet's input audio.

## E.2    OTHER EXPERIMENTAL DETAILS

**Dataset.** Noise was selected randomly from the dataset and repeated up to ten seconds. All audio was sampled single-channel at 16 bits and 16 kHz. All settings had five sources of background noise added to them, with their post-RIR sum being set to various SNRs with respect to the post-RIR clean speech. (While five is a large number of sources, note that $70\,\mathrm{dB}$ SNR still represents an effectively zero-noise setting.)

**Optimal perturbation selection.** Within each trial, the final output was selected as the perturbation with the lowest final loss—i.e., STOI(output, clean) for untargeted attacks, and STOI(output, clean) - STOI(output, target) for targeted attacks. For over-the-air attacks where not every attack iteration converged to the feasible set, we instead selected the final output, as earlier outputs might not satisfy the final masking threshold (see Appendix C).

**STFT.** We use short-time Fourier transforms with Hann windows, 512 FFT points, a window length of 512, and a hop length of 256.

**Optimization Details.** In all experiments except the simulated over-the-air experiment, we parameterize $\delta$ in the time domain. We use Adam for all optimization, using an initial learning rate of 0.01, and clip gradients to a 2-norm of 10. We decrease the learning rate by a factor of 0.99 whenever the loss fails to decrease for 10 iterations in a row.

**Hyperparameter selection.** Our hyperparameters were selected by hand through extensive subjective evaluation (by us) of attack perceptibility and attack success.

**Computation resources.** For all experiments, we used a GPU with at least 40 GB of VRAM (generally an A40, A100, or L40S), 8 CPU cores, and 40 GB of RAM. Except where noted, experiments took less than 2 hours with this setup.

# F  HUMAN STUDY DETAILS

## F.1  DESIGN AND COUNTERBALANCING

*Transcription* comprised 18 clips per participant: for each {model} × {background} × {stimulus type} combination we included one clip, where stimulus types were (**Attacked Input**) noisy+reverberant speech with an imperceptible perturbation, (**Clean Output**) model output for clean input, and (**Attacked Output**) model output for attacked input. To avoid leakage and learning effects, no clip was reused across stimulus types for the same participant. *ABX* comprised 12 pairs per participant constructed from the attacked-input and unattacked-input pools. We also included two ABX attention checks with clearly audible degradations: these followed the same design as the primary samples, except their attacks were generated with a *positive* masking threshold offset of 16.0 (other attacks in the human study used a negative offset of -16.0). Clip identities were fixed across participants to standardize difficulty and reduce computational burden.

The second seed (sample) out of 20 was manually omitted from the study due to being nearly-unintelligible whispering. Otherwise, for simplicity, assignment of seeds to (model, stimulus type, background) configurations was done through iteration using that hierarchy: iterating from 0, 2, 3, ..., 18, we changed background every seed, stimulus type every two seeds, and model every six seeds.

## F.2  STATISTICS

Let $p \in \mathcal{P}$ index participants and $i \in \mathcal{I}$ index items (clips). For the transcription task, let $Y_{pi,c} \in [0,1]$ denote the word accuracy (WAcc) for condition $c \in$ {Attacked Input (AI), Clean Output (CO), Attacked Output (AO)} on the $(p,i)$ cell (observed when that participant transcribed that clip under that condition). Let $\mu_c = \mathbb{E}[Y_{pi,c}]$ be the population mean WAcc for condition $c$.

**Two-way ("pigeonhole") bootstrap for crossed effects.** To obtain CIs that respect crossed participant–item variance (Owen, 2007), we independently resample participants and items with replacement. On each bootstrap replicate $b = 1, \dots, B$, draw counts $U_p^{(b)}$ for $p \in \mathcal{P}$ from $\mathrm{Multinomial}(|\mathcal{P}|; 1/|\mathcal{P}|, \dots)$ and $V_i^{(b)}$ for $i \in \mathcal{I}$ analogously; form replicate weights $W_{pi}^{(b)} = U_p^{(b)} V_i^{(b)}$. For any condition $c$, the replicate mean is

$$\bar{Y}_c^{*(b)} = \frac{\sum_{(p,i) \in \mathcal{D}_c} W_{pi}^{(b)} Y_{pi,c}}{\sum_{(p,i) \in \mathcal{D}_c} W_{pi}^{(b)}},$$

where $\mathcal{D}_c \subseteq \mathcal{P} \times \mathcal{I}$ is the set of observed cells under $c$ (missing cells receive weight 0 by construction). Percentile one-sided $(1 - \alpha)$ CIs use the empirical quantiles of $\{\bar{Y}_c^{*(b)}\}_{b=1}^B$.

**Primary claim via an intersection–union test (IUT).** We test that the attacked output is *simultaneously* less intelligible than both the attacked input and the clean output:

$$H_0 : (\Delta_1 \geq 0) \text{ or } (\Delta_2 \geq 0) \qquad \text{vs} \qquad H_1 : (\Delta_1 < 0) \text{ and } (\Delta_2 < 0),$$

where the mean differences are $\Delta_1 = \mu_{AO} - \mu_{AI}$ and $\Delta_2 = \mu_{AO} - \mu_{CO}$. By the IUT, an overall level-$\alpha$ test rejects $H_0$ iff *each* elementary one-sided test rejects at level $\alpha$. Operationally, using the two-way bootstrap, compute the $(1 - \alpha)$ *upper* percentile bounds

$$U_k = \mathrm{Quantile}_{1-\alpha}\left(\{\Delta_k^{*(b)}\}_{b=1}^B\right), \qquad \Delta_k^{*(b)} = \bar{Y}_{AO}^{*(b)} - \bar{Y}_{(\cdot)}^{*(b)}, \quad k \in \{1,2\},$$

with $(\cdot) = \mathrm{AI}$ for $k = 1$ and $(\cdot) = \mathrm{CO}$ for $k = 2$. Reject $H_0$ (and conclude AO is worse than both) iff $U_1 < 0$ *and* $U_2 < 0$. In our data, the 95% upper bounds were $U_1 = -0.464$ and $U_2 = -0.458$, yielding rejection. Note that, by construction, the IUT does not need a multiple-comparisons correction.

**ABX discrimination.** Let $Z_{pi} \in \{0,1\}$ indicate whether participant $p$ correctly identified the attacked clip in pair $i$. We test $H_0 : \mu_{ABX} \leq 0.5$ vs $H_1 : \mu_{ABX} > 0.5$, where $\mu_{ABX} = \mathbb{E}[Z_{pi}]$. Using the same two-way bootstrap on $\{Z_{pi}\}$, form the *lower* one-sided $(1 - \alpha)$ percentile bound $L$ for $\mu_{ABX}$ and reject $H_0$ iff $L > 0.5$. Our lower 95% bound was $L = 0.478$, so we did not reject.

Table 4: External assets (excludes standard Python libraries installed with `pip`, versioning for which is included in the code's `requirements.txt`).

| Asset | Authorship / citation | License | Version / commit | Special terms of use |
|---|---|---|---|---|
| openai-Whisper | Radford et al. (2022); | MIT | v20240930 / `90db0de` | None beyond MIT |
| MP-SENet | Lu et al. (2023); | MIT | `611cd89` | None beyond MIT |
| Denoiser | Defossez et al. (2020); | MIT | `8afd7c1` | None beyond MIT |
| FRCRN (ClearerVoice-Studio) | Zhao et al. (2022) | Apache-2.0 | `634f004` | None beyond Apache-2.0 |
| MaskGCT (Amphion) | Wang et al. (2024a) | MIT | `f7cb4b4` | None beyond MIT |
| DNS-Challenge (datasets & code) | Dubey et al. (2022); | CC-BY-4.0 (data), MIT (code) | `4dfd2f6` | Attribution required for dataset |
| DNSMOS P.835 | Reddy et al. (2022); same repo/commit as above | CC-BY-4.0 (data), MIT (code) | `4dfd2f6` | Attribution required |
| FullSubNet-Plus | Chen et al. (2022); | Apache-2.0 | `0d11530` | None beyond Apache-2.0 |
| NISQA | Mittag et al. (2021); | MIT | `fe84f0f` | None beyond MIT |
| ViSQOL | Hines et al. (2015); | Apache-2.0 | `b2b2a64` | None beyond Apache-2.0 |
| ViSQOL Docker image | jonashaag/visqol, Docker Hub; https://hub.docker.com/r/jonashaag/visqol | Not declared (container redistributes ViSQOL Apache-2.0) | 'v3' | Used only for build convenience; inherits upstream license |

## G LICENSE DETAILS

See Table 4.

