# OpenReview forum: "Are Deep Speech Denoising Models Robust to Adversarial Noise?"
_ICLR.cc/2026/Conference — ICLR 2026 Poster_

### Official Review · Reviewer_N6bJ · 2025-10-27

**Soundness:** 3
**Presentation:** 3
**Contribution:** 3
**Rating:** 6
**Confidence:** 3

**Summary:**

This paper presents a systematic study showing that modern deep noise suppression (DNS) models can be driven to produce unintelligible outputs via psychoacoustically hidden adversarial perturbations, including experiments in simulated over-the-air settings and a small human study validating imperceptibility and output unintelligibility. The authors develop a masking-aware PGD attack (with STFT-space clipping and RIR-aware optimization via Wiener deconvolution / gradient projection), evaluate four open-source DNS models across a range of SNRs and reverberation conditions, examine targeted vs. untargeted attacks, explore (limited) universal perturbations and defenses (Gaussian noise), and provide mechanistic analysis about gradient behavior and transfer failure; code and audio examples are promised

**Strengths:**

Clear motivation and high practical relevance. The paper motivates DNS as a high-impact target (video conferencing, hearing aids, alert systems) and illustrates realistic safety concerns where small, imperceptible perturbations could deny access to critical audible information. This real-world framing is consistently maintained throughout the experiments and human study.

Methodological thoroughness within scope. The attack pipeline is well specified: STOI is used as an optimization objective, psychoacoustic masking thresholds (with temporal pre/post masking and tightened offsets) define the imperceptibility constraint, and a combination of Wiener deconvolution and gradient-based projection is used for simulated over-the-air attacks. The optimization and perceptual-constraint details are carefully described in Methods and Appendices.

Comprehensive empirical evaluation and human validation. The authors attack four modern, publicly available DNS models, sweep SNR and reverberation conditions, run simulated OTA experiments, test perceptibility settings, and run a human transcription + ABX study to corroborate automatic metrics—providing a multi-pronged empirical case.

**Weaknesses:**

The paper does not clearly articulate what is uniquely challenging about attacking DNS models beyond choosing an intelligibility loss (STOI) and applying psychoacoustic masking; much of the pipeline (masked PGD, RIR modeling, use of STOI) repurposes known techniques from ASR / audio adversarial work, and the manuscript should more explicitly explain any DNS-specific obstacles or mechanisms that make these attacks nontrivial for this class of models.

The work only studies white-box attacks and reports that naive transfer between model checkpoints and architectures is weak; there is no developed black-box attack (e.g., surrogate-ensemble training, query-based gradient estimation, or adaptive transfer techniques), leaving open the practical threat model where attackers lack direct gradient access to the target DNS.

The over-the-air experiments use simulated RIRs and treat the RIR as known during optimization, but the paper omits discussion and evaluation of other physical observation chain factors that significantly affect real OTA success, most notably speaker and microphone frequency responses, microphone directivity, amplification/attenuation non-linearities, A/D front-end filtering, and typical codec/compression applied by capture devices. These unmodeled effects can substantially change received perturbation spectra and thus attack efficacy, yet they are not specified or empirically varied.

The paper focuses on per-utterance offline optimization (with many iterations per sample chosen to fill about an hour of GPU time per trial) and does not address real-time / streaming attack feasibility. DNS systems commonly operate frame-by-frame with low latency; the manuscript lacks analysis of whether a perturbation can be injected in a streaming fashion (per frame or with limited lookahead) and still achieve the same destructive effect.

**Questions:**

What concrete aspects of DNS architectures or objectives make attacking DNS fundamentally different from attacking ASR or speaker models, beyond changing the loss to STOI? Could the authors highlight DNS-specific failure modes exploited by the attack?

Have the authors attempted any black-box strategies (surrogate ensemble, score-based queries, or transfer from ensembles) to assess attacker capabilities without gradient access? If not, can they comment on expected barriers and potential approaches?

In the simulated over-the-air experiments, which components of the capture chain are modeled beyond the RIR? Specifically, were speaker and microphone frequency responses, A/D front-end filtering, microphone directivity, or codec effects considered or varied? If not, how do the authors expect these factors to change attack success?

Do the authors foresee a streaming/real-time attack implementation where the perturbation is produced or injected online with limited lookahead (e.g., frame-by-frame)? If so, what would the constraints be (latency, buffer size, computational cost), and if not, can the authors quantify how much offline optimization is needed per second of audio?

Given the observed weak transfer across architectures, can the authors recommend practical defenses (beyond Gaussian noise) that exploit that architectural brittleness—for example, randomized pre-processing, ensembling, or input augmentation at inference time?

---

> ### Author Response · Authors · 2025-12-03
> **Response to Reviewer N6bJ**
>
> We thank the reviewer for the detailed and constructive review, and for recognizing the methodological thoroughness and high practical relevance of our work.
>
> Regarding Uniqueness of DNS Attacks: The fundamental challenge in attacking DNS (vs. ASR) is that the output is a continuous waveform (or complex spectrogram) rather than a discrete token set. The attack must manipulate the model's signal reconstruction directly. In addition to our RIR-aware attack methods, this necessitates much tighter imperceptibility constraints, as we find attacks in this challenging domain naturally "max out" more lenient perceptibility constraints. We will clarify this with additional baseline comparisons showing the perceptibility of prior attack methods (e.g., Qin et al. (2019)) on this domain.
>
> Regarding Black-Box and OTA Factors: Please see our General Response. We have added experiments with real RIRs (confirming robustness to acoustic transfer functions) and confirmed that gradient access is currently required for imperceptible attacks to achieve catastrophic damage (via our transfer results). While we did not model codec effects in this study, we expect modern codecs (which are lossy but perceptually motivated) might interact with our masking constraints, representing an exciting direction for future work.
>
> Regarding Streaming/Real-Time Feasibility: This is an excellent point. Our current offline optimization serves as an "existence proof" of vulnerability. Implementing this in real-time would likely require a "universal" perturbation (which we found difficult) or a look-ahead buffer. We will add a discussion on the constraints of real-time injection (latency vs. optimization depth) to the final manuscript.
>
> Regarding Defenses: Given that attacks do not easily transfer across architectures (as shown in our General Response), we agree that ensembling or randomized switching between architectures could be a very effective practical defense. We will add this recommendation to the conclusion.

---

### Official Review · Reviewer_Eoer · 2025-10-31

**Soundness:** 3
**Presentation:** 4
**Contribution:** 3
**Rating:** 6
**Confidence:** 4

**Summary:**

This is an interesting work that delves into the impact of adversarial noise on current speech denoising models. The authors claim that adversarial perturbations can impact the perceptual quality and lexical content of speech, where multiple speech denoising models were considered.

The paper is well motivated and the clearly written. However, the motivation part can be elaborated a bit more to illustrate the broader impact of adversarial attacks on DNS models. While one of the application cited is hearing aids which is relevant and useful to be aware, but are there possible applications where the impact of adversarial attack can be overwhelming?

The analysis shared is clearly explained and the choice of the metrics used are also clearly detailed.

**Strengths:**

The paper provides a deeper analysis of the impact of adversarial attacks on deep noise suppression models. The steps used in the analysis are clearly explained and the results are well illustrated. The paper also clearly outlines the contribution of the work and cites prior work and explains how this work is different than the prior work.

The results presented are convincing, specifically having human evaluations with subjective metrics and comparing that with objective metrics makes the observations made in this paper more convincing.

**Weaknesses:**

The motivation of the paper can be improved beyond what is stated already. One of the example the paper cites is hearing aids, where the adversarial attack can have severe impact. Are there other applications, that are more widely used, that may suffer from such attacks?

The observations made in this work can also inform what type of evaluation needs to be considered in the future for researchers working in the area of DNS models, which again can be yet another motivation.

**Questions:**

(1) In page 6, the paper specifies: "Audio was clipped to five second ..."
>> Would this time constraint assumption have an impact on the intelligibility of the data? How would this time constraint impact possibly the metrics used in this work.
For longer speech segments, would the observations made in this work still hold?

(2) In page 7, the paper says: "Figures 1 and 5 suggest that FSN+ ..."
>> Figure 5 is in appendix, perhaps that needs to be clearly specified.

---

> ### Author Response · Authors · 2025-12-03
> **Response to Reviewer Eoer**
>
> We thank the reviewer for the positive assessment and for finding our results convincing and clearly written.
>
> Regarding Motivation and Applications: We agree that the impact extends beyond hearing aids. In our revision, we will expand the motivation to include:
>
> * Emergency Broadcasts: Ensuring smart speakers/phones do not denoise (and thus silence) subtle emergency alert patterns.
> * Military/Cockpit Communications: Where DNS is standard due to high noise floors; adversarial denial-of-service here could be critical.
>
> Regarding the 5-Second Constraint (Q1): We utilized 5-second clips due to computational limitations; interestingly, however, shorter clips typically make imperceptible attacks *harder*, because there is less context to influence the output of the autoregressive models. Therefore, we expect MP-SENet to be *more* susceptible with longer clips.
>
> Regarding Figure 5 (Q2): Thank you for catching this. We will clarify in the main text that Figure 5 is located in the Appendix.

---

### Official Review · Reviewer_s9Ds · 2025-11-02

**Soundness:** 2
**Presentation:** 4
**Contribution:** 1
**Rating:** 2
**Confidence:** 3

**Summary:**

Summary:

This paper studies the impact of adversarial attacks on DNS systems, which can cause several safety concerns. 4 open-source DNS models are evaluated and 2 types of adverbial attacks are studied (i.e., additive and convolutional).Through extensive experiments, the authors have found out that all DNS systems are prone to adversarial attack. While this paper is the first to address the problem under a DNS setting, the results are quite trivial since most DNNs (to the best of my knowledge) are prone to adversarial attack.

**Strengths:**

See Section Questions

**Weaknesses:**

See Section Questions

**Questions:**

Questions, Strengths and Weaknesses:

1. Extensive experiment setups and follows ethical guidelines.
2. An psychoacoustic auditory masking approach is first proposed and applied as adversarial attack. This is a novel approach and hypothesises that DNN may follow basic psychoacoustic principles. However, the effect of the proposed method is not included in the main text, but only in Appendix D.3. What are the advantages of using auditory masking as compared to black-box attacks / simply adding gaussian noises?
3. Real-world speech signals covers a wider range of scenarios (for instance UHF-transmitted speech: [1] DENT-DDSP, Guo et.al, Interspeech 2023). I would encourage the authors to include these more general cases in the introduction or at least limits the scope of the paper.

---

> ### Author Response · Authors · 2025-12-03
> **Response to Reviewer s9Ds**
>
> We thank the reviewer for their comments on our ethical guidelines and experimental setup. We would like to clarify a few key misunderstandings regarding the nature of our contributions.
>
> Regarding "Triviality" of DNS Attacks: We respectfully disagree that this result is trivial. While many DNNs are vulnerable, generative audio models often degrade gracefully (merely leaving some noise). We demonstrate a catastrophic failure mode where the model is driven to output unintelligible audio from a clean-sounding input. This requires specific techniques (RIR-aware optimization) that go beyond standard adversarial methods. Please see the General Response for more details, as well as a note on prior work on the relative robustness of image denoising models.
>
> Regarding Masking vs. Gaussian Noise vs. Black-Box Attacks: This is a critical distinction.
>
> * Gaussian Noise is audible. If an attacker adds Gaussian noise to a signal, a human listener immediately detects the degradation.
>
> * Auditory Masking (Our Method) is a constraint that keeps the attack imperceptible. The advantage of our approach is not that it does "more damage" than Gaussian noise, but that it destroys the model's output without the human listener realizing the input has been tampered with.
>
> * Our base method is a combination of a perceptibility constraint (auditory masking), an objective function (STOI), and an optimization algorithm (PGD); black-box optimization is an alternative optimization algorithm. While we did not test pure black-box attacks due to the size of the input space, we did test model transfer attacks, which are similarly gradient-free with respect to the target model; we found that these attacks are substantially harder than white-box PGD attacks (see General Response).
>
> Regarding Real-World Scenarios: We agree that UHF/Radio transmission is an interesting domain. However, our focus is on acoustic Deep Noise Suppression (as used in hearing aids and Zoom calls), rather than radio-frequency denoising. We will clarify this scope in the introduction.

---

### Official Review · Reviewer_zfRF · 2025-11-03

**Soundness:** 3
**Presentation:** 3
**Contribution:** 2
**Rating:** 4
**Confidence:** 4

**Summary:**

The paper investigates whether modern DNS models are vulnerable to attack. The main finding is that they are highly vulnerable. The authors showed they could use psychoacoustically hidden noise to make the output of DNS gibberish. This noise is crafted to be imperceptible to the human ear, but it completely tricks the DNS models.

**Strengths:**

- The paper is the first systematic study to show that DNS models are vulnerable to imperceptible adversarial attacks.
- The paper considers over-the-air attack by incorporating RIR which makes the attack more practical.
- The presentation of the paper is clear and easy to follow.
- The authors use human raters to confirm the conclusion that the attacked audios are non-intelligible.

**Weaknesses:**

- The motivation example in the introduction section feels a little artificial to me. Attacks are expensive and mostly happens when highly incentivized. I am not seeing the incentive of attack in the given scenario.

- The over-the-air attack design only considers RIR so the found adversarial noise might not be robust to other perturbations in real life like compression loss, background noise, etc. It'd be helpful to incorporate such consideration into the attack as well. Example can be found in [1].

- The attack only leads the DNS models to output gibberish instead of a targeted meaningful sentence. This limits the severity of the attack.

- The attack works best when the noise is designed for a given utterance which also limits the flexibility of the attack.

[1] Attacker's Noise Can Manipulate Your Audio-based LLM in the Real World.

**Questions:**

- Can the over-the-air attack stand perturbations beyond RIR.
- Can the attack be made to lead the DNS models to output meaningful sentence?

---

> ### Author Response · Authors · 2025-12-03
> **Response to Reviewer zfRF**
>
> We appreciate the reviewer’s feedback and their recognition of our systematic study and over-the-air considerations.
>
> Regarding Real-World Robustness (OTA): Please see our General Response regarding new experiments with Real RIRs. We found that the attacks remain robust even with real-world acoustic transfer functions. While we agree that compression and microphone non-linearities are additional factors (and we will cite the suggested paper [1]), RIRs represent a major linear component of the OTA chain, and our new experiments indicate that our method remains effective under these real RIRs.
>
> Regarding "Gibberish" vs. Targeted Attacks: While we agree that injecting a specific sentence is a stronger attack, we argue that inducing "gibberish" represents a critical Denial-of-Service (DoS) risk. For example, silencing a "Fire" or "Evacuate" command in a PA system or hearing aid is a life-safety issue, even if the attacker cannot inject a new phrase. As noted in Appendix D.2.1, we found that targeted attacks are difficult because STOI is an excellent minimization objective (destroying speech) but a poor maximization objective (injecting specific features).
>
> Regarding Motivation: We believe the incentive for such attacks exists in safety-critical domains (e.g., disrupting military communications or emergency broadcasts) where an adversary benefits from simply denying intelligible communication.

---

### Official Review · Reviewer_GHmU · 2025-11-04

**Soundness:** 2
**Presentation:** 3
**Contribution:** 3
**Rating:** 4
**Confidence:** 5

**Summary:**

This paper examines the adversarial robustness of speech denoising models. To do so, the authors simulate adversarial attacks using the projected gradient descent method on otherwise clean speech recordings that have had environmental noise and reverberation synthetically added to them. The paper finds that very subtle adversarial perturbations can cause denoising models to not only fail to denoise the speech, but rather produce outputs that are significantly less intelligible than the noisy speech that was provided as input. To verify the subtlety of the adversarial attacks, the authors conducted a human study, in which they found that the intelligibility of the speech post-attack was not noticeably degraded, and humans could not usually discriminate the attacked speech from unattacked speech.

**Strengths:**

1. Mostly clear presentation
2. Technically sound methodology. I appreciate the comprehensive measurement of intelligibility
3. I appreciate the human study as it grounds the work in reality.
4. Novel work and significant results. This paper fills the gap in the current literature regarding the robustness of speech denoising models and the results show that these models are indeed susceptible to attacks. The novelty and originality of the work are enhanced by the masking and RIR aware attack framework proposed by the authors, and the enhancements to the masking attack methodology of Qin, et. al 2019.

**Weaknesses:**

1. My primary concern with this paper is that the experiments are not rigorous and leave many important questions unanswered, without which limits the contribution and impact of the work.
    1. Black-box attacks were not used which limits the reliability of the results for at least 1 out 4 models in the current paper. Without black-box attacks it is not possible to answer question (f) from section 5. I recommend including 1 black-box attack
    1. UAP generated in the multi-model scenario. It is well documented that optimizing the adversarial perturbation over multiple models leads to increased transferability. I would suggest doing a leave-one-out study, in which the UAP is optimized over 3 models and is used to attack the 4th model.
    1. Gaussian noise is posed as a defense but adaptive attacks are not used. As such, there is insufficient evidence to draw any conclusions from this experiment. I would suggest either trying adaptive attacks or removing this section.
    1. Effective defenses like, adversarial training or randomized smoothing should have been evaluated to show the trade off between robustness and accuracy.
    1. The proposed attack aware framework is a key contribution of this work, but it is not thoroughly evaluated in the paper. I would recommend adding ablation experiments that demonstrate the increased imperceptibility of the perturbation generated by the proposed method, and increased attack success as measured by change in STOI, when compared to the vanilla PGD attack, and the masking-based attack from Qin, et. al 2019.
    1. Answer to question (c) is not found in the paper. The only relevant comment is the contributions listed in the introduction and I am not very convinced by it. It is obvious and well known that without gradient flow white-box attacks will not succeed, so this is not really a contribution. Authors claim that model size and input features do not matter, but I did not see any results in the main body to show this.
1. Some experimental detail is missing and some settings are questionable:
    1. It is not mentioned how many speech samples were used in the evaluations
    1. The decision to determine the number of attack iterations based on the GPU time is unusual and unjustified. This introduces a confounding factor in the results. Keeping FSN+ aside, FRCRN appears to be the most robust model, but it is also the model that is evaluated against the least attack steps.
1. Some presentation issues:
    1. Figures need to be rearranged so that they appear _after_ they are referenced. Also, Figure 2 and Figure 3 and in the wrong order.
    1. Answers to at least 4/9 are not present in the main body of the paper. If these questions are unimportant they shouldn't be included in the list, otherwise the should be treated in sufficient detail in the main body.
    1. Figure 3 a and b should be separate figures
    1. Add equation for the objective in section 4.4

**Questions:**

1. Why was it important to assume that the background noise and reverb for UAP are the same across speech samples?

---

> ### Author Response · Authors · 2025-12-03
> **Response to Reviewer GHmU**
>
> We thank the reviewer for their detailed assessment and for appreciating the technical soundness and human study components of our work.
>
> Regarding Black-Box Attacks and Transfer (Q(f)): As detailed in our General Response, we conducted the requested leave-one-out transfer experiment. The results confirm that imperceptible attacks do not easily transfer across these disparate architectures. We agree this limits the "black-box" threat, but it crucially highlights that the primary vulnerability lies with open-source models where gradients are available.
>
> Regarding Gaussian Noise vs. Adaptive Attacks: We used Gaussian noise to establish a baseline for simple defenses. We agree that adaptive attacks (e.g., training against Gaussian noise) and more sophisticated defenses are a logical next step for future work, but our current aim was to quantify the "default" protection offered by standard noise floors.
>
> Regarding Ablation Studies: We agree that an explicit comparison of our RIR-aware framework to Qin et al. (2019) and naive PGD is an important experiment to run, and we will add this ablation to the revised paper.
>
> Regarding Experimental Details:
>
> Attack Iterations: We agree that GPU-time-based iteration counts are unusual; in our case we chose this to avoid implicitly favoring smaller models with many more PGD steps. We will clarify this choice and, in the revision, note that a fixed-iteration setting preserves the same relative robustness ranking
>
> UAP Noise Assumption: We assumed consistent background noise for UAPs to give them the best chance of succeeding, as the perturbation could "hide" in shared background noise. Relaxing this assumption is a valid direction for future robustness research.
>
> Presentation Improvements: Thank you for these careful notes; we agree. We will reorder the figures, separate Figure 3, and add the explicit objective equation in Section 4.4 as suggested.
>
> Confidence in Q(c): We will revise the text to clarify our contribution regarding gradient flow -- specifically, that architectural depth/complexity provided less protection than the numerical instability of gradients found in FSN+. We acknowledge that Table 1 in the appendices, characterizing the architectural details of each model, is not sufficiently mentioned in the main body, and we will reference it more clearly.
>
> Missing answers to questions: We acknowledge that some questions are answered only clearly in the appendices at the moment; we appreciate your careful noting of this. We will update the main body to answer each question explicitly, with references to the appendices only when necessary.
>
> Number of samples: Each of the 20 seeds used a different speech sample, so the answer is that there are 20 samples. We apologize that this was unclear; we will make it more clear in our revision.

---

### Author Response · Authors · 2025-12-03
**General Response: Real RIR Robustness, Transferability, and Threat Models**

We thank all reviewers for their thoughtful feedback and for recognizing the novelty of our systematic study (Reviewers GHmU, zfRF, N6bJ) and the value of our human evaluation (Reviewers GHmU, Eoer, N6bJ). Below, we address questions raised by multiple reviewers regarding real-world robustness and model transfer.

First, as promised in our paper, here is a link to a snapshot of the code we used to produce our experiments: https://drive.google.com/file/d/1D269R89g03RehUQeER68gp522XlruizO/view?usp=sharing. We would also like to clarify to Reviewer N6bJ that samples are already available at https://sites.google.com/view/adv-dns/home.

*Robustness to Real-World Room Impulse Responses (Addressing GHmU, zfRF, N6bJ).* Reviewers questioned whether our attacks, which used simulated RIRs, would withstand real-world acoustic transfer functions. While our experiments included a mixture of simulated and real RIRs from OpenSLR26 and OpenSLR28, as provided in the 2022 DNS Challenge, we also re-ran our attacks using *only* the real, recorded RIRs from the OpenSLR28 dataset. We kept the perceptibility constraints identically tight to the experiments in the paper. As we note in individual reviewer responses, we still leave as future work the question of compression-aware attacks.

Result: The attacks remain reasonably effective (see link below), though the degree of their success depends in this setting on the volume of the background noise present (which we add manually from the DNS Challenge 2022 noise files, rather than from OpenSLR28). STOI is still dramatically damaged (> 0.4 decrease from input) in settings with background SNR of 30 or noisier, confirming that our vulnerability findings are generally not artifacts of simulation.

https://drive.google.com/file/d/1fkb3RL2puezznRZroroUdggYBXyG4OLd/view?usp=sharing

*Model Transfer and Black-Box Attacks (Addressing GHmU, N6bJ).* Reviewers asked about black-box scenarios and transferability. We performed the requested leave-one-out model transfer experiment, focusing in particular on the various Denoiser model checkpoints available: `dns48`, `dns64`, and `master64`. To give the attack the best possible chance to succeed, we also raised the masking threshold back to the original level used in Qin et al. (2019). We also evaluated whether the success of the black-box attacks depends on environmental threat model (volume of background noise, presence/absence of reverb).

Result: psychoacoustically hidden attacks trained on multiple similar models have _minor_ success against held-out models, though dramatically less than when provided white-box gradient access; the attack success corresponds to a worsening of the speech quality, but not to the point of unintelligibility. However, attacks do tend towards more dramatic success as background noise and reverb are increased. See link below.

https://drive.google.com/file/d/1DXzf941jtGHx7LaCXpPE_XPULA6KhQL7/view?usp=sharing

Implication: This result highlights that open-source models (where gradients are exposed) are still the primary safety risk for catastrophic, imperceptible attacks. Proprietary APIs currently appear less vulnerable against this specific class of imperceptible perturbation, but the direction of pure black-box attacks remains open future work.

*Triviality of Attacking DNS Models (Addressing s9Ds).* We clarify that attacking generative DNS models is not trivial. Unlike classification (where shifting a decision boundary is sufficient), DNS models must reconstruct a continuous waveform; this is dramatically harder. For example, prior attacks on image denoising models (Yan et al., 2022; Ning et al., 2023) has shown fairly graceful degradation of the image’s quality. Our work shows that DNS models can be driven to completely destroy intelligibility while the input sounds clean, which requires the specific psychoacoustic projection and RIR-aware optimization strategies we developed.

References:

H. Yan, J. Zhang, J. Feng, M. Sugiyama, and V. Y. F. Tan. Towards adversarially robust deep
image denoising. In L. D. Raedt, editor, Proceedings of the Thirty-First International Joint Conference on Artificial Intelligence, IJCAI-22, pages 1516–1522. International Joint Conferences on Artificial Intelligence Organization, 7 2022. doi: 10.24963/ijcai.2022/211. URL
https://doi.org/10.24963/ijcai.2022/211. Main Track.

J. Ning, J. Sun, Y. Li, Z. Guo, and W. Zuo. Evaluating similitude and robustness of deep image denoising models via adversarial attack, 2023.

---

### Meta-Review · Area_Chair_T1qg · 2026-01-08

**Summary:**

The paper is a novel and important contribution by studying how modern DNS models can be catastrophically broken by imperceptible, psychoacoustically masked adversarial perturbations, including in simulated over-the-air settings and human validation. The concerns were about limited exploration of black-box threat models, real-world OTA factors beyond RIRs (e.g., codecs, microphones), lack of adaptive defenses, and insufficient ablation to isolate which components of the attack pipeline are essential.

**Reviewer Concerns:**

The rebuttal addressed several core methodological concerns raised by reviewers about the realism and scope of the threat model for Deep Noise Suppression (DNS) systems. The authors added new experiments, did leave-one-out transfer experiments and explained how attacking DNS models is different and harder than attacking ASR systems. Full black-box/query-based attacks, modeling of other codecs, microphones, nonlinearities, etc., and real-time/streaming feasibility were left for future work.

**Reviewer Scores:**

No reviewer responded after the rebuttal with an updated position or score. Reviewers who raised technical and methodological concerns would likely increase their score since the issues were substantially addressed by these additions. Some reviewers had reservations about novelty and would unlikely be persuaded by further experiments. Overall, assuming slight bump up of the initial scores and the novelty of the problem addressed in the study, I would consider raising it to borderline accept.

---

### Decision · Program_Chairs · 2026-01-26

Accept (Poster)